# Steps dominate gas evasion from a mountain headwater stream

Gianluca Botter [1] ✉, Anna Carozzani[1], Paolo Peruzzo [1] & Nicola Durighetto[1]

Steps are dominant morphologic traits of high-energy streams, where climatically- and biogeochemically-relevant gases are processed, transported to downstream ecosystems or released into the atmosphere. Yet, capturing the imprint of the small-scale morphological complexity of channel forms on large-scale river outgassing represents a fundamental unresolved challenge. Here, we combine theoretical and experimental approaches to assess the contribution of localized steps to the gas evasion from river networks. The framework was applied to a representative, 1 km-long mountain reach in Italy, where carbon dioxide concentration drops across several steps and a reference segment without steps were measured under different hydrologic conditions. Our results indicate that local steps lead the reach-scale outgassing, especially for high and low discharges. These findings suggest that steps are key missing components of existing scaling laws used for the assessment of gas fluxes across water-air interfaces. Therefore, global evasion from rivers may differ substantially from previously reported estimates.

River networks transport, process, and release a multitude of chemical substances, which are relevant to the biogeochemical functioning of stream ecosystems and eventually affect the fragile interconnections of the land-water-climate nexus[1–4]. In particular, headwater streams are important greenhouse gas sources to the atmosphere, owing to the combination of enhanced input of dissolved matter from the surrounding landscape with high exchange rates across water-air interfaces. Consequently, quantifying gas emissions from upland freshwater systems is regarded as an important scientific challenge with multi-faced implications for a broad range of disciplines including ecology, biology, and climate sciences[5–8].

Riffles represent a distinctive trait of most rivers worldwide[9]. In high-gradient streams, where the granulometry is varied, riffles frequently give rise to sequences of steps in which local hydraulic discontinuities of the water flow are observed. Step bedforms include a lot of incredibly diverse channel structures, which are widespread in different regions of the globe, including humid areas, desert ephemeral streams, semiarid environments, and alpine settings[10,11]. In some instances, the local morphology of the stream jointly with the enhanced turbulence observed in correspondence with the plunging jet may promote the formation of a submerged pool, where the characteristic travel time of water and solutes increases significantly. The important role of steps and step-pool bedforms in regulating fluvial sediment transport and channel morphology has been extensively studied[12–19], but systematic knowledge about their contribution to gas exchange between freshwater systems and the atmosphere remains elusive.

Several authors have argued that waterfalls, riffles, steps, and cascades might promote gas exchange with the atmosphere, owing to the enhanced turbulence and air entrainment that are typically observed in correspondence of abrupt discontinuities of the flow field[7,20–28]. However, available empirical data about the outgassing produced by individual steps or cascades is relatively limited. Cirpka et al.[20] and Natchimuthu et al.[29] have used tracer injections to demonstrate that the presence of cascades and waterfalls significantly increases the reaeration coefficient in a set of tens-of-meters long river reaches located in Switzerland and Sweden. More recently, Leibowitz et al.[30], Vautier et al.[31] Whitmore et al.[32] and Schneider et al.[33] have shown that mass evasion within channel stretches that contain waterfalls is enhanced, causing a loss of carbon dioxide ($CO_2$) or injected tracers within relatively short distances in the range of 15% to 50% of the initial mass. However, spatial patterns of gas evasion were

[1]Department of Civil, Environmental and Architectural Engineering, University of Padua, via Marzolo 9, 35131 Padua (PD), Italy.
✉e-mail: gianluca.botter@dicea.unipd.it

typically monitored at relatively coarse spatial resolutions (i.e. some tens of meters), and empirical observations at scales comparable to the step size are much rarer (see ref. [31]). High-resolution data, instead, represent a powerful means to reduce the uncertainty in the characterization of small-scale patterns of stream outgassing, and enable more robust assessments of gas evasion produced by local hydromorphologic heterogeneities of rivers[34].

Existing studies aimed at quantifying the relative contribution of cascades to the total stream outgassing typically rely on spatial patterns of gas concentration within river reaches that contain steps or waterfalls[30,31,33]. These patterns, however, inherently mirror the unique morphologic characteristics of the case studies selected for the analysis. Consequently, the existing estimates can not be easily upscaled or extrapolated to different contexts. In other instances, gas evasion produced by steps and cascades was investigated through the analysis of the underlying mass transfer rate, $k$[28,29,31,35]. Though, the mean value of $k$ in a given stream portion is not able to quantify the magnitude of the internal peaks of gas transfer occurring in the case of non-homogeneous hydrodynamic fields[36], especially when the size of the fluid volume responsible for most of the evasion is significantly smaller than the measurement resolution—as in the presence of a falling jet (Supplementary Fig. 1). Thus, the value of $k$ in correspondence of steps, riffles, cascades, and waterfalls is highly scale dependent (the larger the averaging water volume around the jet, the lower the corresponding mean value of $k$). For this reason, we suggest that the mass transfer rate might not be an appropriate metric to describe gas exchange processes in correspondence with abrupt discontinuities of the flow field, where the energy dissipation is markedly heterogeneous and the actual water volume involved in the majority of gas evasion is unknown (and potentially very small).

Owing to these theoretical and practical limitations—in spite of the growing awareness of the importance of local heterogeneity of the flow field in water-air gas exchange—the relative contribution of steps to the total outgassing in morphologically-complex reaches is not fully clear. This study aims at filling this gap by developing a theoretical and experimental framework for the study of gas emissions in heterogeneous streams. The approach was applied to a high-gradient channel in the Italian Alps where water $CO_2$ concentrations were measured across 19 natural and artificially created steps, and along a reference turbulent segment without steps (Fig. 1). The main innovation of the experimental setup is that we capitalize on direct $CO_2$ records gathered under different discharge conditions, without relying on Schmidt-number scaling, which is problematic in case of bubble-mediated transport of the type observed downstream of steps and cascades[26,35,37,38]. The major theoretical advance, instead, pertains the development of a modular and scalable metric for disentangling the contribution of turbulent segments and local steps to the total stream outgassing. The scalability of this metric was exploited to assess the role of local steps in the reach-scale outgassing, taking into account both the temporal variations of the discharge and the morphological characteristics of the river bed.

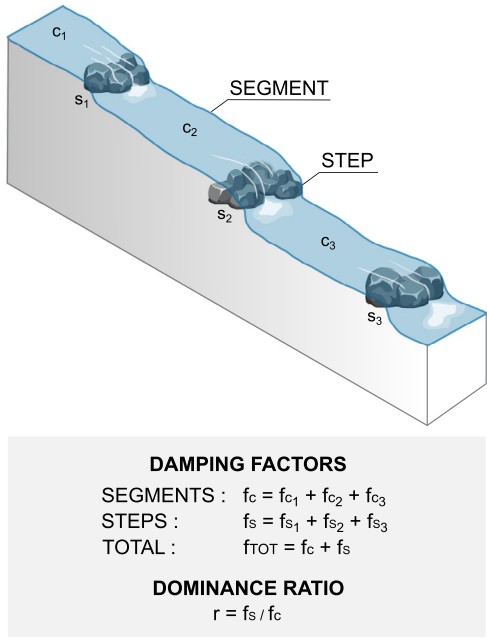

**Fig. 2 | Schematic of the decomposition of a reach into segments and steps.** Definition and aggregation of the damping factors $f_C$ and $f_S$ within a complex system, which leads to the expression of the dominance ratio $r$. Image generated with Inkscape.

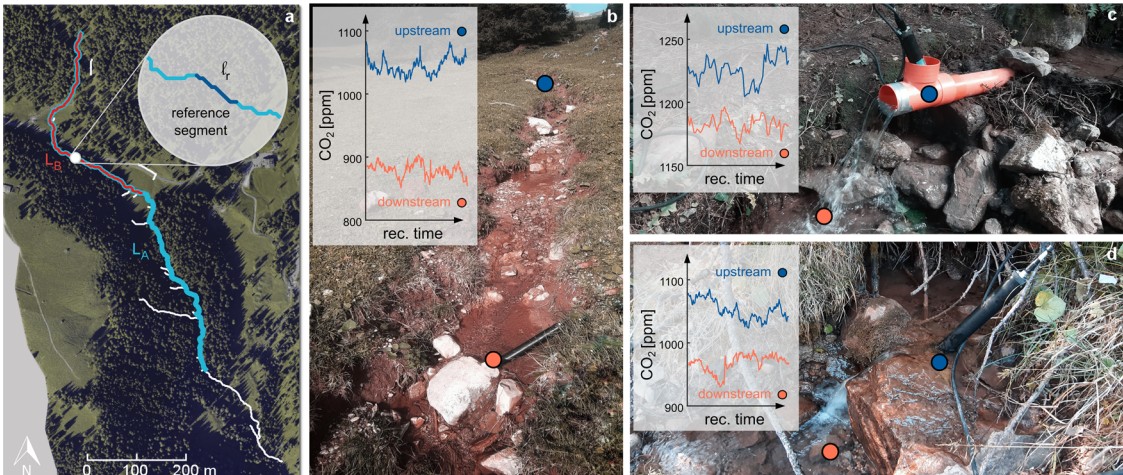

**Fig. 1 | Focus reach of the Valfredda creek and experimental setup. a** planar view of the reach selected for this study, shown with an orthophoto of the eastern part of the Valfredda catchment in background. Here, light blue and red indicate reach A, with length $L_A = 1060$ m, and reach B, with length $L_B = 543$ m, respectively. The inset within the planar view shows the reference segment without steps of length $\ell_r = 13$ m. **b** overview of the reference segment indicated in (**a**). **c** Example of an artificial step, created by forcing the stream into a pipe and then covering the downstream river bed with a plastic film. **d** example of a scoured natural step. The insets in the last three panels show the observed time series of water $CO_2$ concentration in the upstream (blue line) and downstream (orange line) cross sections, the positions of which are indicated as blue and orange circles in the three pictures.

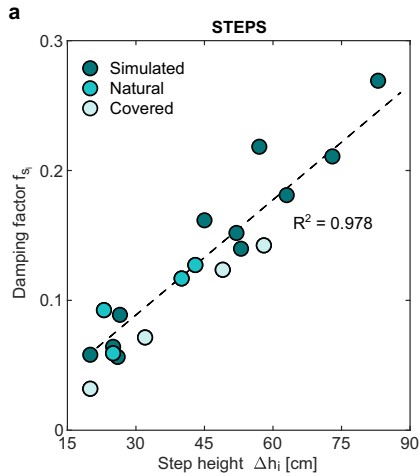

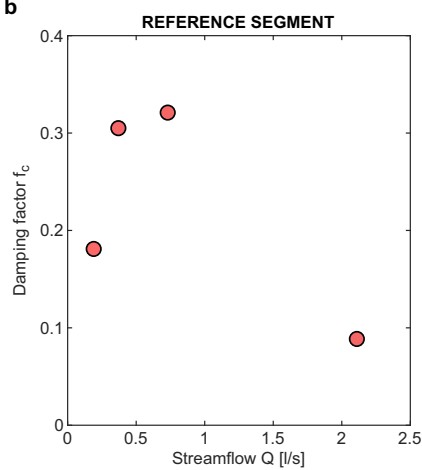

**Fig. 3 | Damping factors for the steps and the reference segment. a** damping factor of the step $i$, $f_{s_i}$, as a function of the step height $\Delta h_i$ for scoured natural steps (cyan circles), natural covered steps (celeste circles) and artificially simulated steps (green circles). The dashed line represents the linear relation $f_{s_i} = 0.3 \Delta h_i$ (95% CI of slope: 0.27, 0.32), with $\Delta h_i$ in $m$, which was obtained by fitting a simple linear regression through the least squares approach on the data ($n = 19$, $p$-value < 0.001). $R^2 = 0.978$ is the coefficient of determination of the linear regression. **b** damping factor in the reference segment, $f_{c_r}$, for different discharge conditions in the range from 0.19 to 2.11 $l/s$.

## Results and discussion

### Concentration damping in streams

For the sake of simplicity, we conceptualize high-gradient stream networks as a heterogeneous sequence of two types of elements: steps and segments (Fig. 2). Local steps are point-wise hydraulic discontinuities of the flow, generated by a drop of the riverbed $\Delta h$ higher than the typical flow depth ($\Delta h > 10$ cm in this case). In such circumstances, the presence of a falling jet promotes air entrainment, bubbles and foaming which enhance gas exchange with the atmosphere. Turbulent segments, instead, are continuous, relatively regular river stretches located between pairs of steps, in which the flow is gradually varied. Therein, turbulence and gas exchange are promoted by heterogeneities of the velocity field, which are in turn produced by e.g. hurdles, stones, bends, and bed roughness. In cases where a geomorphic pool is observed downstream of a step, the pool is considered to be part of the downstream segment—provided that the outgassing process generated by a step is highly localized around the falling jet, regardless of the presence of the pool (Supplementary Fig. 1).

Evaluating the role of local steps in river gas evasion requires the definition of a metric capable of objectively determining the separate contribution to the outgassing of steps and turbulent segments. To this aim, we use the concept of concentration damping, which is a dimensionless, scalable measure of gas evasion applicable to individual steps, single segments, or composite heterogeneous channels. Under steady-state conditions, the 1$D$ spatial pattern of the concentration $C$ of a dissolved gas advected downstream (x direction) and evaded into the atmosphere is exponentially decreasing with a spatially heterogeneous decay rate. Therefore, $C(x)$, i.e. the concentration in the position $x$ along the streamline, can be expressed as a function of the concentration in the upstream section ($x = 0$), $C_0$, and the atmospheric concentration $C_a$ as

$$C(x) - C_a = (C_0 - C_a) \exp[-f(x)] \qquad (1)$$

where $f(x)$ is the exponential damping factor (hereafter damping factor), defined as the product between the effective exchange rate along the stretch $(0, x)$, $K_{eq}(x)$, and the corresponding water travel time $\tau(x)$ (i.e. $f(x) = K_{eq}(x)\tau(x)$, see "Methods"). Physically, $f(x)$ provides a measure of the fraction of excess mass (i.e. the mass exceeding that transported by the stream at the equilibrium, when $C = C_a$) removed along the streamline from 0 to $x$, which can be in fact calculated

as $1 - e^{-f(x)}$ ("Methods"). The damping factor of a continuous channel segment (say, $c_i$) of length $\ell_i$ can be written as

$$f_{c_i}(\ell_i) = K_{c_i} \tau_{c_i}, \qquad (2)$$

where $\tau_{c_i} = \int_0^{\ell_i} 1/u(x)\mathrm{d}x$ is the travel time spent by water parcels in the segment (measurable via tracer experiments) and $K_{c_i}$ the effective exchange rate therein.

Analogously, the damping factor of the step $s_i$, $f_{s_i}$, can be expressed as

$$f_{s_i}(\Delta h_i) = K_{s_i} \tau_{s_i}, \qquad (3)$$

where $K_{s_i}$ is the effective exchange rate within the step $i$ and $\tau_{s_i}$ the corresponding travel time. The notation emphasizes that $f_{s_i}$ should depend on the step height $\Delta h_i$, which drives the amount of energy dissipated by the water flow and the ensuing outgassing process. In the case of steps, it is practically unfeasible to separately measure $\tau_{s_i}$ and $K_{s_i}$. Nevertheless, step exchange rates ($K_{s_i}$) are expected to be very high owing to the local increase of energy dissipation and the enhanced air entrainment in correspondence of the falling jet[20,39–41]. Consequently, $f_{s_i}$ could be similar to (or even larger than) $f_{c_i}$ in many settings, in spite of the local nature of the outgassing process in correspondence of the steps (i.e., $\tau_{s_i} < \tau_{c_i}$).

The damping factor is additive and commutative ("Methods"), thereby implying that the value of $f$ for a series of segments (steps) can be calculated as the sum of the damping factors associated with the individual segments (steps) involved, regardless of their specific order (Supplementary Text 1.2). Consequently, we can evaluate the relative contribution to the total outgassing induced by all the steps embedded in a focus reach (or river network) using the dominance ratio $r$, which is defined as the ratio between the damping factor of the steps ($f_s$) and the damping factor of the segments ($f_c$) of that reach (see Fig. 2 and Supplementary Text 1.2):

$$r = \frac{\sum_i f_{s_i}}{\sum_i f_{c_i}} = \frac{f_s}{f_c}. \qquad (4)$$

In Eq. (4), $f_c$ ($f_s$) is expressed as the sum of the damping factors of all the segments (steps) included in the reach (Fig. 2 and "Methods"). If the dominance ratio is equal to unity, then the steps and

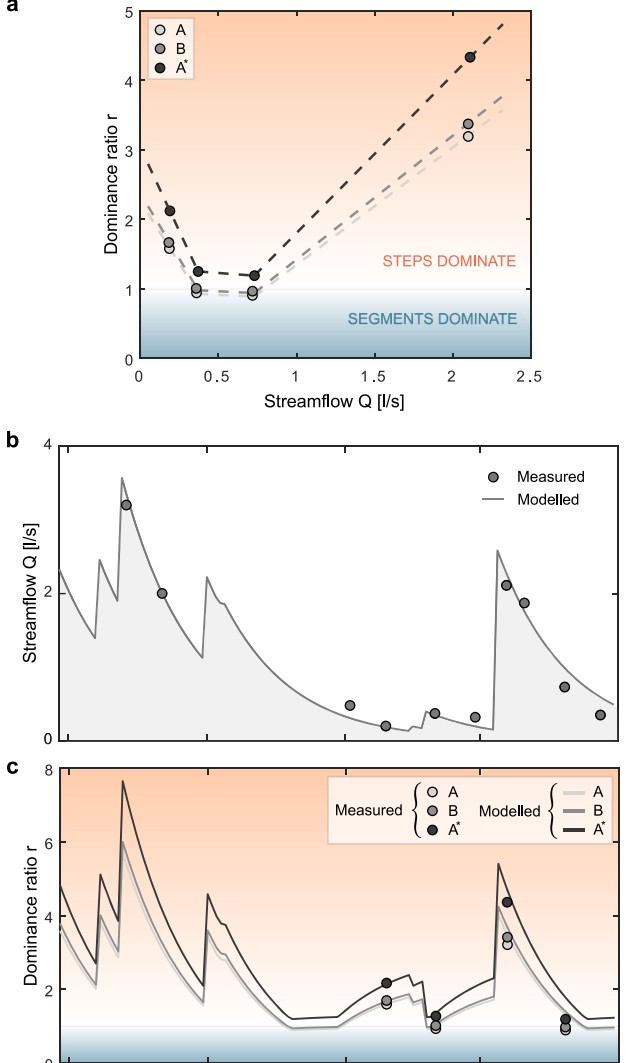

**Fig. 4 | Dominance ratio variations induced by changes in the underlying hydrologic conditions. a** dominance ratio as a function of the stream discharge $Q$ in the reaches $A$ (light gray), $B$ (gray) and $A^*$ (black). Circles represent the observed values of $r$, the dashed lines show a linear piece-wise interpolation. When $r > 1$, steps dominate the outgassing (upper region of the plot). **b** Temporal dynamics of the discharge in the reference segment of the focus reach from July 1 to Nov 1, 2021. Observed streamflows are indicated as gray dots, while the simulated discharges (see "Methods") are shown by the shaded-gray region. **c** Temporal dynamics of the dominance ratio during the same time window (shown in **b**), for reaches $A$ (light gray), $B$ (gray) and $A^*$ (black). Circles refer to the observed values, while the solid lines refer to simulated values estimated based on the simulated discharges (shown in **b**) using the piece-wise linear interpolation between $r$ and $Q$ (shown in **a**). When $r > 1$ the steps dominate the outgassing (upper region of the plot).

the continuous segments of the focus reach provide an equal contribution to the total river outgassing (meaning that the gas concentration damping produced by a sequence of steps would be the same as the concentration damping produced by the turbulent continuous segments without the steps). Likewise, if $r > 1$ then the steps proved a larger contribution to the outgassing—or if $r < 1$ a smaller outgassing—as compared to the contribution generated by the continuous turbulent stretches. Crucially, $r$ is not affected by the specific order according to which segments and steps are arranged, but only depends on key geometrical and hydraulic features of river networks (e.g. step height/spacing, segment slope).

## Steps dominate reach-scale gas evasion

Our experimental setup allowed robust estimates of the $CO_2$ fluxes and damping factors across the steps and the reference segment belonging to the focus reach. Observed concentrations in the focus reach ranged from 823 to 1297 ppm (mean concentration = 1130 ppm, standard deviation = 250 ppm) and the corresponding $CO_2$ fluxes released into the atmosphere were comprised between 0.49 and 15.4 $gC/d$ for the steps, and between 1.0 and 7.1 $gC/d$ for the segment (Supplementary Tables 4, 6). In the reference segment, the damping factor $f_{c_r}$ ranged from 0.1 to 0.32, depending on the underlying flow rate, $Q$ (Fig. 3). These values fall within the range of damping factors observed in the literature (Supplementary Fig. 11), and corresponded to normalized gas exchange rates, $k_{600}$, in the interval (3–15) $m/d$, which led to a percentage of excess mass released into the atmosphere (calculated as $1 - e^{-f_{c_r}}$) between 10% and 25%. The non monotonous dependence of $f_{c_r}$ on $Q$ was explained by the interplay between two important drivers of gas transfer across water-air interfaces: i) the mean flow velocity, which is positively correlated with the outgassing velocity and is an increasing function of $Q$ (Supplementary Fig. 5); and ii) the ratio between exchange area and water volume, which is proportional to the exchange rate but decreased with $Q$ in the reference segment (Supplementary Table 3). Consequently, $f_{c_r}$ peaked for intermediate discharges.

In the range of drop heights analyzed in this paper (from 0.2 to 0.83 m), the step damping factor $f_s$ varied between 0.03 and 0.27, depending on the underlying elevation drop $\Delta h$. As the discharges observed in the study reach were quite small (<3 $l/s$), we also included in our analysis a scoured natural step of the main Valfredda river, with an height of 0.43 $m$ and a discharge much higher than that observed in the focus reach (108 $l/s$). In all cases $f_{s_i}$ had the same order of magnitude of $f_{c_r}$, demonstrating that a local step may originate nearly the same outgassing produced by turbulent river segments with a length of tens of meters. This similarity is also reflected by the values of the $CO_2$ fluxes released by the steps into the atmosphere, which were comparable to those evaded along the reference segment (Supplementary Tables 4, 6).

Interestingly, $f_{s_i}$ exhibited an almost linear dependence on the step height $\Delta h_i$ and was statistically independent on the discharge $Q$ (Fig. 3a and Supplementary Tables 5, 6). In particular, the natural step characterized by a discharge exceeding 100 $l/s$ perfectly lies along the linear regression line of all points. The above evidence indicates that gas exchange in steps might not be primarily related to the turbulence generated by the jet impacting the free surface, which is known to increase with $Q$. Instead, we hypothesize that $f_s$ is likely to be driven by bubble-mediated processes—the magnitude of which might be controlled by the total jet energy, which is a linearly increasing function of $\Delta h$.

The natural steps, where $CO_2$ production was impeded by manually scouring and removing all the biofilm from the river bed and the downstream pool prior to each measurement, showed a behavior which was essentially indistinguishable from that of the artificially created steps, suggesting that the procedure used for simulating the steps with pipe diversions did not introduce significant biases in our analysis. The dependence of $f_{s_i}$ on the step height ($\Delta h_i$) in natural steps covered with the plastic film was in line with that observed for the other steps, while the underlying absolute values were slightly lower—likely because in this setting the falling jet adhered to the plastic film, thereby reducing the reaeration rate and the gas evasion in the covered steps.

The ratios between $f_s$ and $f_c$ in the three target reaches analyzed in this study (the reaches $A$ and $B$, and the virtual reach $A^*$, see "Methods"), were in the range [0.9–4.3], depending on the underlying discharge level and the specific reach analyzed. These values were calculated taking into account the number and heights of the steps within each reach, and using a linear empirical function to link $f_{s_i}$ to $\Delta h_i$ as suggested by our experimental data (dashed line in Fig. 3a). Interestingly, most of the gas evasion

induced by the steps was associated with the smallest drop heights. In fact, about 60% of $f_s$ was contributed by steps with heights smaller than 35 cm (Supplementary Table 8). Provided that $f_s$ turned out to be independent on $Q$ while $f_c$ peaked for intermediate discharge levels, $r$ had a non monotonic dependence on $Q$, with higher values observed for low and high streamflow conditions. While only few experimental points were used to estimate the dependence of the dominance ratio on the discharge, in all the settings analyzed $r$ was systematically larger than (or close to) unity, thereby indicating that the contribution provided by the steps to the total gas evasion from the Valfredda creek was at least 50%, with even higher percentages in correspondence of low-flow and high-flow conditions.

The above estimates of $r$ relied on field measurements which were gathered through four specific surveys, during which the discharge in the reference segment varied between 0.19 and 2.11 $l/s$. Therefore, to understand whether these measurements were representative of the long-term behavior of the system in the entire study period, we analyzed how daily variations in the streamflow drained by the focus reach −driven by fluctuations of rainfall and soil moisture content in the root zone−could impact the temporal variability of the dominance ratio. To this aim, the dynamics of $Q$ during the summer and fall of 2021 were reconstructed combining field measurements and a simple hydrological model. The temporal pattern of $r$ was then reconstructed from simulated discharge variations ("Methods"), using the empirical relationship between $r$ and $Q$ shown in Fig. 4a. Our results clearly indicate that $r$ remained consistently above unity during the whole monitoring season for the three target river reaches analyzed. The highest values of $r$ were observed during high-flow conditions. The resulting average values of the dominance ratio during the whole monitoring period were equal to 2.4 (for reach $A$), 2.5 (for reach $B$) and 3.2 (for reach $A^*$). These values differ from the simple algebraic average of the experimental points shown in Fig. 4a, since they properly take into account the dependence of $r$ on $Q$ and the relative frequency associated with different discharge levels in the focus reach during the study period. We conclude that the outgassing from the study reach of the Valfredda was largely dominated by the local evasion induced by step and pools during the summer and early fall of 2021.

## Implications for large-scale studies

On practical grounds, objectively decomposing morphologically-complex channels of the type investigated here into segments and steps may not be straightforward. Here, steps were identified in correspondence with sharp drops in the active river bed, with heights that exceeded 10 cm. In our experimental setup, drops of this type gave rise to aerated falling jets followed by bubbles and/or foaming in the downstream pool, thereby enhancing gas exchange with the atmosphere. In some cases, however, vertical drops could be very small or involve only a portion of the active riverbed, creating spatially heterogeneous hydrodynamic conditions which are quite difficult to describe. These hybrid elements were not considered as actual steps in this paper−and their contribution to gas emissions was neglected accordingly. Furthermore, owing to the spatial heterogeneity of key hydro-morphological characteristics such as slope, discharge and riverbed composition/roughness, the continuous emissions in all the segments belonging to the study reach might not be perfectly represented by the behavior of the reference segment as postulated by the upscaling procedure proposed in this paper. Nevertheless, in spite of all these limitations and the inherently local nature of our study, we believe that the results shown here can be of general validity and properly describe the order of magnitude of the processes involved. In our study reach, we detected 271 steps in 1.03 km, with a mean distance between two subsequent steps of about 3.8 m, and a mean step height of 42.8 cm (Supplementary Table 7). These numbers are in line with previously published data in other regions of the World. In fact, the mean step spacing was found to be 2.56 m in the Western Cascades

(Oregon, USA)[42], 5.29 m in Southern California[10] and between 3.9 and 6.5 m in Northern Italy[43,44]. Likewise, the mean step height was found to be between 0.47 to 1 m in D'Agostino and Lenzi[43], 0.49 m in Wilcox et al.[44] and 0.22 m in Chartrand et al.[45]. Therefore, the important contribution of small-height steps to the total outgassing observed in the Valfredda is expected to emerge in many high-slope settings, where step and pool bedforms dominate the channel morphology.

Our study revealed that the footprint of gas emissions produced by local steps does not vanish at the reach scale, owing to the pronounced concentration damping in correspondence of each step and the high frequency of steps typical of steep mountain rivers. Although we might expect the specific value of $r$ to be spatially variable from site to site depending on local morphologic and hydraulic features, our analysis provides important clues for the identification of the drivers of $f_s$ and $r$ in river networks and enables the identification of general guidelines for the application of the proposed framework to other contexts. The empirical data collected in this study indicate that $f_s$ could be easily extrapolated to any setting in which steps are observed, as the damping factor seems to be only dependent on the step height $\Delta h_i$. In fact, we did not notice sizable differences among the behavior of the steps created with artificial pipes, that of the small natural steps contained in our target reach (width < 0.5 m, $Q < 3$ $l/s$) and the outgassing of a natural step with a larger width and a much higher discharge (width >2 m, $Q > 100$ l/s). While we recognize that more experimental data gathered under a broader range of conditions would be necessary to make stronger claims, we propose that, for a given step height, the damping factor is nearly the same regardless of other important hydro-morphologic features (step shape, presence/absence of pools, width, discharge). Interestingly, owing to the the additivity of $f_{s_i}$ within a reach with multiple steps and the linear dependence of $f_{s,i}$ on $\Delta h_i$, the number and size of individual steps does not impact the value of $f_s$ of a reach, which instead depends only on the total elevation drop lost through all the steps contained therein, $\Delta h_s = \sum_i \Delta h_i$.

Extrapolating $f_c$ across different segments, instead, is likely to be less straightforward. Several empirical equations taken from the literature could be used for this purpose, in particular the experimental relationship observed between the mass transfer rate and the turbulent kinetic energy dissipation rate, $\varepsilon$[26]. This relation postulates that $f_c$ would be nearly the same in river segments in which the flow velocity and the slope are the same. Therefore, $r$ could be nearly constant across different reaches in which (i) the discharge and velocity are similar; (ii) the mean slope is the same; iii) the fraction of elevation drop taking place through the step is the same. As a consequence, to extrapolate the value of $r$ in a river network, specific data about the small-scale morphological traits of reaches would be necessary, unless this information is surrogated by empirical geomorphic laws for the prediction of the frequency and height of local steps based on larger-scale terrain attributes (see e.g. ref. [45]). Yet, more experimental data gathered within streams of larger size would be necessary to substantiate the proposed method and confirm its suitability to be extrapolated across different scales and settings.

The damping factors and the dominance ratio quantify the potential outgassing of different stream elements, whereas the actual value of evaded mass does depend on the spatial correlation between the sources of matter along the stream and the spatial patterns of evasion[25]. For instance, if relatively high water $CO_2$ concentrations (e.g. induced by external supply of matter from the surrounding hillslopes, hyporheic exchange, and pronounced ecosystem respiration) are observed in those portions of a reach where the value of $f$ is higher (e.g. because the elevation drop induced by the steps is larger), carbon dioxide evasion is expected to be particularly enhanced. Given the local nature of gas emissions from steps, cascades, and waterfalls, these geomorphic elements could act as important emission hotspots, where the excess mass transported downstream by the flow is quickly released into

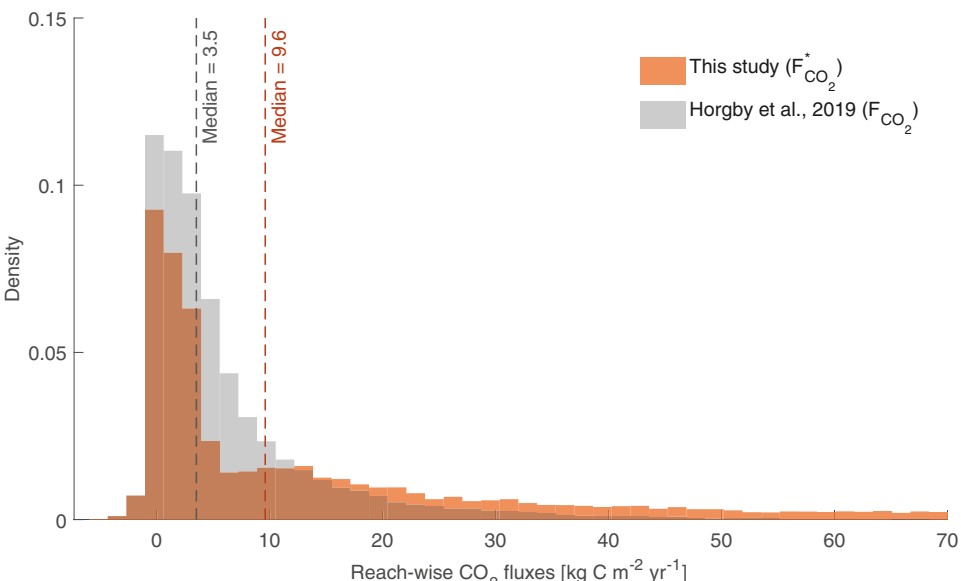

**Fig. 5 | Effect of steps on CO₂ emissions from Swiss mountain streams.** Frequency distribution of reach-wise $CO_2$ fluxes estimated by Horgby et al.[8], $F_{CO_2}$ (gray histograms), and the corresponding frequency distribution of the fluxes estimated by taking into account the local emissions generated by steps, $F^*_{CO_2}$ (orange histograms), for 23,343 Swiss mountain streams. The black and orange dashed lines represent the median flux values estimated by Horgby et al.[8] and this study, respectively. Note that the tail of the frequency distribution including the steps reaches values up to $F^*_{CO_2} \approx 500$ kgC m⁻²yr⁻¹.

the atmosphere (e.g. refs. 23, 27, 30). Carbon dioxide emissions from local steps can be particularly significant if the stream bed in correspondence with these steps is partly exposed and thus covered by biofilm, an instance which is known to enhance local CO₂ production[46,47]. Therefore, there could be a significant amount of outgassed mass in correspondence of steps, which could be essentially undetectable because of the very short distances traveled from the input (or production) site to the evasion point. These CO₂ fluxes from rivers to the atmosphere might not be captured by simplified approaches in which the representative stream CO₂ concentrations are estimated exploiting sparse point-wise measurements[3,5–8,22,24,25,27,29]. More broadly, we propose that the accuracy of current methods that indirectly estimate the stream metabolism through observed gas concentration differentials[48–50] can be highly sensitive to the specific position of the selected sampling points and the small-scale geomorphic characteristics of the corresponding upstream reaches.

The analyses presented in this paper highlight a series of potential shortcomings in the methods currently in use for large-scale estimates of stream outgassing. Steps represent crucial morphologic components of high-energy streams, as they regulate physical or chemical exchanges at the interface with the landscape and the atmosphere. Consequently, a proper characterization of the spatial frequency and height of such steps is an important prerequisite for a robust assessment of gas evasion from channel networks. In heterogeneous high-energy streams, in fact, different stretches characterized by the same mean slope and velocity could lead to highly variable gas evasion rates depending on their internal configuration. For instance, according to Eqs. (8) and (10), the 13 m reference segment considered in this study was able to evade—through a total elevation drop of 1.4 m—approximately 8 % to 27 % of the available excess mass, depending on the underlying discharge rate ($f_c = 0.09$ for $Q = 2.11\,l/s \rightarrow 1 - e^{-f_c} = 0.08$; $f_c = 0.32$ for $Q = 0.73\,l/s \rightarrow 1 - e^{-f_c} = 0.27$). If the same mean slope and elevation drop were obtained by combining nearly horizontal segments with two steps of height 70 cm each, the damping factor would be independent on $Q$ and much higher than that observed in the reference segment (from 30 % to 250 % higher, depending on the underlying discharge value), with a percentage of excess mass evaded

close to 35 % ($f = 0.42 \rightarrow 1 - e^{-f} = 0.35$). The example indicates that for a given mean slope and velocity of a stream—i.e., for a fixed value of turbulent kinetic energy dissipation rate—both the apparent gas exchange rate at the water-air interface and the corresponding gas fluxes could be highly variable depending on the internal configuration of the reach (e.g. the frequency and height of local steps). In particular, higher emissions are expected to be associated with settings in which a sizable proportion of the total elevation drop takes place through local steps, where the steps dominate the outgassing process. Instead, the hydromorphologic parameters used for the prediction of gas exchange rates in ungauged sites (e.g., discharge, mean slope, mean velocity)[5,26,51–53] do not explicitly incorporate the effect of local steps and small-scale heterogeneity in the stream morphology. Therefore, in order to improve the precision of large-scale estimates of gas fluxes from rivers we should develop novel, more sophisticated scaling laws that differentiate between channels with or without steps, linking the apparent gas exchange rate and the damping factor to the frequency and height of the steps contained in a reach.

The theoretical innovation introduced by this paper might imply a paradigm shift in gas emission studies, redirecting the efforts of the scientific community toward a better characterization of small-scale heterogeneity of streams and the development of more integrative approaches able to overcome the limits of the mass transfer rate—a metric which is not suited to describe local processes such as the outgassing in correspondence of the falling jet of steps, cascades, and waterfalls. The direct implications of our findings for large-scale assessments of gas evasion are certainly relevant. On the one hand, if existing studies have disregarded the effect of steps the actual flux of greenhouse gases across water-air interfaces of mountain streams would be much larger than what currently foreseen in the literature. This might be the case if previous empirical studies had been preferentially performed in stream reaches where the continuous segments dominate over the steps. To demonstrate the important role of steps for large-scale gas emissions from river networks, we revised the estimate of the flux of $CO_2$ released from Swiss mountain streams performed by Horgby et al.[8]—that does not account for local steps—and we extended that calculation to include the step contribution to gas evasion in high-energy rivers. To this aim, the dominance ratio $r$ was calculated for each reach of the Swiss

network as a function of the slope (as higher slopes imply more frequent and higher steps, thereby leading to higher $r$) and discharge (higher $Q$ imply higher flow velocities and higher mass transfer rates, with lower values of $r$). Although the above extrapolation goes beyond the range of discharges and slopes observed in the present study and thus should be taken with extreme caution, our calculations indicated that the estimate of the flux of $CO_2$ released from Swiss mountain streams given by Horgby et al.[8] would need to be corrected from approximately 3.5 to about 9.5 g $CO_2$/m²/y (Fig. 5) if the steps are accounted for. In particular, we observed a significant increase in the number of reaches where the evasion is larger than 15 g $CO_2$/m²/y owing to the emissions from the steps embedded in the steepest branches of the Swiss network. These results hint once again at the key role of local morphological traits of rivers for global emissions of $CO_2$ into the atmosphere. On the other hand, in the case in which the effect of steps was already (at least partly) included in existing large-scale estimates—simply because some of the high energy experimental reaches used for the development of scaling laws do contain steps—quantifying the large-scale impact of steps on gas emissions from rivers would be even more troublesome. In this case, amending current estimates of regional $CO_2$ fluxes would require not only a systematic characterization of the morphological traits of the reaches where mass transfer rates were previously measured, but also the identification of potential biases in extrapolating available data across ungauged reaches. In fact, the upscaling procedures currently in use lack of a suitable stratification based on key step features (in particular, the fraction of height drop associated with local steps), which seems to be instead a necessary step forward in stream outgassing studies. A literature analysis revealed that data about the internal structure of the river reaches used for experimental tracer studies—lying at the basis of large-scale predictions of gas emissions—are seldom available. Thus, we propose that more efforts are needed to collect and analyze data about the small-scale geometry of streams where gas evasion has been measured. Better characterizing the local morphological traits of streams that regulate the mass exchanged through water-air interfaces could help us to constrain the budget of focal chemical species (e.g. carbon, oxygen, nitrogen) relevant to the water-land-climate system.

## Methods

### Damping factor

The equation governing the spatial patterns of gas concentration in a one-dimensional system with a curvilinear coordinate $x$ aligned with the main flow direction, under the assumptions of stationarity (constant flow rate $Q$ and time-invariant gas concentrations), no dispersion, no lateral input, absence of internal gas production reads:

$$u(x)\frac{dC(x)}{dx} + K(x)[C(x) - C_a] = 0 \,, \tag{5}$$

where $u(x)$ is the local velocity in the streamline direction, $C_a$ the atmospheric concentration, $C(x)$ the local water gas concentration, $K(x)$ the local, spatially variable exchange rate, which is equal to the mass transfer rate $k$ scaled to the mean water depth. Crucially, the exchange coefficient $K$ embeds the coupled effects of the mass transfer induced by the turbulence of the flow and that associated with gas transport mediated by bubbles and foams (if any, see ref. 54). The solution of Eq. (5) is given by

$$C(x) = C_a + (C_0 - C_a)\exp\left[-\int_0^x \frac{K(x')}{u(x')}dx'\right] = C_a + (C_0 - C_a)\exp[-f(x)] \tag{6}$$

In Eq. (6), the exponential damping factor $f(x)$ is defined as

$$f(x) = \int_0^x \frac{K(x')}{u(x')}dx' = \int_0^{\tau(x)} K(t')dt' = K_{eq}(x)\tau(x), \tag{7}$$

where $x'$ is the integration variable (representing any arbitrary position between 0 and $x$), $K_{eq}(x)$ is a weighted spatial average of $K$ in the stretch from 0 to $x$ and $\tau(x) = \int_0^x 1/u(x)dx$ is the corresponding transit time—the time necessary to travel from 0 to $x$. Manipulating both sides of Eq. (1), one can easily derive the following expression for the mass removed in $(0, x)$ scaled to the excess mass $Q(C_0{-}C_a)$ (i.e., the maximum value of mass that can be removed before the equilibrium with the atmosphere is reached):

$$\frac{Q(C(x) - C_0)}{Q(C_0 - C_a)} = 1 - e^{-f(x)} \tag{8}$$

Thanks to Eq. (1), $f_c$—the damping factor of an ideal channel stretch of length $L$ composed by all the continuous segments of the focus reach—can be expressed as

$$f_c = \ln\left[\frac{C_0 - C_a}{C_L - C_a}\right], \tag{9}$$

where $C_0$ and $C_L$ are the gas concentrations in the upstream ($x = 0$) and downstream ($x = L$) sections of the reach. Operationally, given the practical impossibility of measuring $f_{c_i}$ within all the segments belonging to the focus reach, $f_c$ was calculated based on the value of the damping factor of a reference segment (see below) with length $\ell_r$, $f_{c_r}$. The latter was estimated from Eq. (9) through direct gas concentration measurements as

$$f_{c_r} = \ln\left[\frac{C_0 - C_a}{C_{\ell_r} - C_a}\right], \tag{10}$$

where $C_0$ and $C_{\ell_r}$ are the concentrations in the upstream and downstream sections of the reference segment. Then, $f_c$ was calculated from $f_{c_r}$ as

$$f_c = f_{c_r}\frac{L}{\ell_r}, \tag{11}$$

exploiting the additivity of $f_{c_i}$ across multiple segments and assuming that the exchange rate in the reference segment is equal to the average value of $K$ across all the segments contained in the focus reach (Supplementary Text 1.6).

Similarly, $f_{s_i}$ was calculated from Eq. (1) as

$$f_{s_i} = \ln\left[\frac{C_{u_i} - C_a}{C_{d_i} - C_a}\right], \tag{12}$$

where $C_{u_i}$ and $C_{d_i}$ represent the water gas concentration upstream and downstream of the step $i$. The damping factor of a sequence of $N$ steps was then evaluated summing up the damping factors of all the individual steps $f_{s_i}$ (i.e., $f_s = \sum_i f_{s_i}$, see Supplementary Information), as in Eq. (4) of the main text.

### Study site and focus reach

The study site selected in this paper is a step-pool channel of the Rio Valfredda, a high-gradient headwater catchment of the Piave river basin, in the Italian Alps[36,55]. The climate of the site is typically alpine: precipitation is relatively high throughout the year (annual rainfall > 1400 mm), with significant snowfall during winter and melting in spring[56]. The selected reach is 1.36 km long, and its elevation ranges

from 1911 to 1720 m a.s.l., with a mean slope of 0.14 m/m (Fig. 1a). The reach was selected because of its accessibility and the significant $CO_2$ concentrations observed therein (typically above 1000 ppm). The river bed is steeper in the upstream part, where it flows southwards. Then, the reach runs south-east across some pastures and a mixed larch-spur forest. The reach is fed at its source by a groundwater spring, and the pH ranges between 7.6 and 8.1. The discharge weakly increases downstream, owing to the interplay between the losing bed and the hillslope lateral input. The stream bed is silty and dominated by boulders, cobbles and wooden logs of different size that originated several steps and pools. About 300 $m$ upstream of its confluence with the Valfredda, the channel was almost inaccessible due to the presence of a landslide and several fallen trunks. Therefore, the analysis was concentrated in the upper portion of the channel (reach $A$ in Fig. 1a).

## Study design

The idea behind this study is to evaluate the contribution of steps and segments to the total gas evasion in streams by measuring $CO_2$ concentration drops within continuous segments and individual steps belonging to our study reach. However, the use of $CO_2$ concentration time series to quantify the outgassing of different stream elements required the confounding effect of $CO_2$ production/input to be eliminated. This goal was achieved by isolating the water flowing across segments and steps from the river bed using a plastic film. The reach $A$ was decomposed into 270 segments and 271 steps (Fig. 1b). Therein, during the summer of 2021 water $CO_2$ concentrations were measured upstream and downstream of 19 different steps with variable height and a reference segment with a slope (discharge) similar to the mean slope (mean discharge) of the continuous segments within the study reach. This reference segment was identified in the middle part of the reach (46°22'50"N, 11°49'39"E, see Fig. 1a) with the aim of quantitatively representing the continuous gas emissions from all the segments contained in the study reach. This reference segment has an average slope $i_{c,r}$ of 0.108 m/m and a length $\ell_r$ of 13 $m$ (inset of Fig. 1a, b). $CO_2$ concentration measurements were performed under different hydrologic conditions (i.e. variable discharges) and considering different types of steps with heterogeneous geometry, as detailed in the following sections of the "Methods".

## Discharge measurements

We performed several volumetric measurements of the discharge rate, $Q$, at the two end points of the reference segment. This was done recording the filling time of a graduated tank in correspondence of the upstream and downstream section of the reference segment. We performed 10 measurements between July and October 2021, with observed discharge values between 0.2 and 3.2 l/s (see Supplementary Table 1). Each measurement was the average among at least 5 different replicas performed in both the locations within one hour.

## Travel time measurements and estimation of the relevant hydraulic properties

The water travel time along the reference segment, $\tau_{\ell_r}$, was measured through instantaneous injections of a diluted sodium chloride (NaCl) solution in the upstream cross section of the segment. The temporal variations of specific conductivity at the outlet of the segment were then measured using a multi-parameter sonde (YSI EXO2). The travel time was recorded both under natural conditions and after having covered the stream bed with the plastic film (Supplementary Fig. 3 and 4). The longitudinal mean velocity, $u$, was estimated as the segment length, $\ell_r$, divided by the observed travel time. The procedure allowed us to verify that the mean velocity along the reference segment was not significantly impacted by the presence of the plastic film. Moreover, after having covered the river bed with the plastic film, we also measured the mean width, $W$, and water depth, $H$ of the flow. This was done

by taking spatial averages of the local values observed in different cross sections along the segment. The obtained hydraulic geometry scaling relationships (Supplementary Fig. 5) were found to be in line with the relationships proposed in the literature for mountain streams[8], thereby suggesting that the flow conditions in place during the $CO_2$ measurements were nearly-natural.

## $CO_2$ concentration measurements

Paired upstream and downstream $CO_2$ concentration measurements were taken in the reference segment and in 19 steps (see Supplementary Text 1.5), using a membrane-based NDIR sensor, the MiniCO$_2$™ designed by ProOceanus Systems Inc., Bridgewater, Canada. The instrument has a tubular shape, 370 mm long and 53.4 mm in diameter, and uses infrared detection to measure the partial pressure of dissolved $CO_2$. Once the internal gas is fully equilibrated with the surrounding water (typically 10–15 min after the deployment), NDIR measurement on the equilibrated internal gas is taken at a wavelength of 4.26 $\mu m$ close to the absorption band of $CO_2$ at a controlled optical cell temperature. The time of deployment in a given position ranged from 30 to 90 min. To eliminate the confounding effect of high-frequency fluctuations in the recorded $CO_2$ signal of the MiniCO$_2$™ sensor (see Fig. 1 for an example), at least 20 min of continuous measurements at steady-state were gathered, from which we estimated the probability density function of $CO_2$ concentrations and the related mean. Steady-state conditions were pre-identified based on the temporal patterns of the long-term average of the signal. The steady-state mean was then taken as representative of the equilibrium carbon dioxide concentration in water (Supplementary Figs. 10, 12). Paired upstream-downstream concentrations were gathered within 120 min from each other, so as to reduce as much as possible spurious effects induced by diel variations of water $CO_2$ concentration. In our analysis we neglected possible effects of $pH$ spatial variations on observed $CO_2$ concentrations, as the underlying $pH$ spatial gradients across the segment and the steps were below the detection limit of our instrument (which was 0.1 for the multi parametric sonde used in this study). Atmospheric $CO_2$ concentrations were also measured with our MiniCO$_2$™ sensor after each measurement performed upstream or downstream of the segment and the steps. $C_a$ was in the range between 390 and 412 ppm throughout the field campaign. These values were in substantial agreement with those recorded in the nearest stations of the World Data Center for Greenhouse gases (Monte Cimone, Sonnblick Observatory, Zugspitze). At any rate, the impact of small variations in the value of $C_a$ on the main paper results was negligible.

## Estimating $f_{s_i}$ and $f_{c_r}$

In this study the damping factors were evaluated on a purely experimental basis. The damping factor of individual steps $f_{s_i}$ was estimated from upstream/downstream $CO_2$ measurements via Eq. (12), considering three different step types: (i) 11 simulated steps, which were created forcing the water into pipes and then letting the water flow hit a covered portion of channel bed from a given height, so as to reproduce the behavior of a falling jet of a natural drop with the desired $\Delta h$ (Fig. 1c and Supplementary Fig. 6); (ii) 4 covered steps, obtained by folding natural steps with a thin plastic film, which was carefully shaped around the actual channel bed in the ramp, the step and the downstream pool (Supplementary Fig. 7e); (iii) 4 natural steps belonging to the focus reach (Fig. 1d and Supplementary Fig. 7f), which were scoured to remove the existing biofilm prior to each measurement. These precautions allowed us to eliminate the effect of $CO_2$ production in all the analyzed steps, while assuring natural hydraulic conditions in all the measured steps. Further details on the experimental setup are available in Supplementary Text 1.5.

As per the estimate of the damping factor in the reach segments, owing to the practical impossibility to quantify the outgassing within all the segments contained in the study reach, $f_c$ was estimated from

the observed concentration drop in the reference segment by means of an upscaling procedure (Eq. (11)). To measure $f_{c_r}$, prior to each field measurement the stream bed was covered by a plastic film, to avoid lateral input of water and $CO_2$ and internal production induced by the ecosystem metabolism (Supplementary Fig. 9). Then upstream vs downstream $CO_2$ concentrations were measured under different hydrologic conditions (discharge range: from 0.19 to 2.11 $l/s$), and Eq. (10) was applied. In the light of the constraint placed by the specific slope of the reference segment and the mean slope of the segments contained in the reach $A$, we analyzed three different scenarios, in which the damping factor of the reference segment was upscaled via Eq. (11) referred to three different target reaches: (i) the whole reach $A$, which has a length $L_A$ of 1060 m and a mean slope of its continuous segments $i_{c,A}$ of 0.081 m/m; because $i_{c,A}$ is significantly smaller than the slope of $\ell_r$, in this case $f_c$ should be overestimated; (ii) the reach $B$, including 130 steps, which has a length $L_B$ equal to 543 m and is characterized by an average slope of its continuous segments $i_{c,B} = 0.101$ m/m—quite close to the slope of the representative segment; (iii) the reach $A^*$, an idealized reach characterized by (i) the same elevation drop as that observed in between the two end points of reach $A$; and ii) segments that have the same slope of the reference segment. $A^*$ is 769 m long, has a slope of its segments equal to 0.108 m/m (by definition equal to $i_{c,r}$) and contains all the 271 steps of reach $A$ (further details in Supplementary Text 1.6). Since the mean slope of the segments included in $B$ and $A^*$ is closer to the actual slope of the reference segment, the corresponding estimates of $f_c$ and $r$ should be more reliable in this case. Note that the mean slope of the segments included in a reach was calculated starting from the mean slope of the overall reach, taking into account the heights of all the steps included in that reach and assuming a longitudinal size of 10 cm for each step (Supplementary Text 1.6). This implies that the length of the segments included in each target reach is slightly smaller than the length of the whole target reach (Supplementary Table 7).

### Morphological survey

The dominance ratio depends on the spatial frequency and the height distribution of the steps in the focus reach. We collected geometrical data about the step geometry during field surveys performed under very dry conditions. For each step we measured the step drop height, $\Delta h$, corresponding to the gap of water surface elevation in each nearly vertical fall with a drop higher than 10 $cm$. We mapped 271 steps in 1060 m with an average $\Delta h$ equal 23.7 cm (Supplementary Fig. 13). The frequency distribution of the step height was monotonically decreasing, with 47.6 % of steps in the range 0−15 cm, and 70.5% of steps in the range 0−25 cm. Elevations, lengths and slopes of the relevant reaches were estimated through a high resolution (1 m) $DTM$.

### Streamflow regime

The streamflow regime was estimated from rainfall data using a simple rainfall-runoff model. In particular, we used daily precipitation heights $P$ [mm] recorded during the summer and fall seasons (June to October 2021) collected by a weather station of the Veneto Region Environmental Protection Agency (ARPAV) located in Falcade, 4.5 km far away from the catchment centroid. Discharge time series in the focus reach were then simulated using an exponential IUH applied to the censored precipitation time series as

$$Q(t) = A \sum_i j_i\, k\, e^{-k(t-t_i)} \tag{13}$$

where $A$ is the catchment area, $k$ is the recession rate, $t_i$ is the occurrence time of effective rain events, $j_i$ is the effective (i.e. censored) rain depth ($j_i = max(P_i - \phi, 0)$, with $P_i$ representing the total rain depth and $\phi$ the censoring threshold embedding soil moisture dynamics[57]. $\phi$ and $k$

were calibrated against the discharge observations available (Fig. 4c and Supplementary Text 1.8).

### Estimation of $CO_2$ emissions from Swiss mountain streams

$CO_2$ fluxes released from from the mountain streams of the entire Switzerland were estimated by integrating the procedure identified by Horgby et al.[8] with a simplified method to include the effect of gas evasion from local steps. As in Horgby et al.[8], we considered the stream network provided by the Swiss Federal Office for the Environment, and we associated with any individual reach of the network the corresponding value of mean annual discharge (reference period: 1981–2000) (FOEN, 2016[58]). For each reach of the network, the stream length ($L$) and the mean slope ($i$) were then calculated using a 2 m digital elevation model (Geodata© swisstopo[59]). Likewise, stream width ($W$), water depth ($H$) and water velocity ($u$) were estimated starting from the underlying mean discharge exploiting three scaling laws developed by ref. [9], in which the values assumed by the empirical parameters involved were taken from Horgby et al.[8]. The procedure indicated by Horgby et al.[8] was then used to estimate areal fluxes from each continuous stream segment, $F_{CO_2}$, from simulated reach-scale values of $CO_2$ concentration and mass transfer rate $k$ (for more details on the data and the methods the reader is referred to ref. [8]). Assuming that these values of $F_{CO_2}$ do not include the effect of local steps, as stated by the authors of the study, one can estimate the value of $f_c$ for each reach of the Swiss network from the available spatial map of $k$. The latter estimate was performed using equation (2), in which $K$ was estimated as $k/H$ and $\tau$ was set equal to $L/u$. Then, the estimate of $F_{CO_2}$ provided by Horgby et al.[8] was amended by adding the contribution to the outgassing produced by local steps, assigning to any reach of the Swiss network a total damping factor equal to $f_{TOT} = f_c + f_s = f_c(1+r)$ (instead of simply $f_c$). To estimate the value of $f_s$ pertaining to a given reach, owing to the linearity of the relationship between $f_{s,i}$ and $\Delta h_i$, only the total elevation drop associated with the steps of that reach, $\Delta h_s$, needs to be known. Owing to the lack of detailed morphological data, $\Delta h_s$ was estimated for each reach of the Swiss network from the mean step spacing, $\lambda_s$, and the mean step height, $\overline{\Delta h}$. The latters were in turn calculated as a function of the slope and the width of the reach as $\lambda_s = 0.3113\, i^{-1.188}$ and $\overline{\Delta h} = i\, W$, exploiting geomorphic relationships taken from the literature[45,60]. Then, the total elevation drop associated with steps in a reach of length $L$ was calculated as $\Delta h_s = (L/\lambda_s)\, \overline{\Delta h}$ and $f_s$ was estimated as $f_s = 0.3\, \Delta h_s$ (see Fig. 3a). Note that the steps were assumed to be relevant only in the stream reaches where $\overline{\Delta h} > 0.5\, H$—otherwise we assumed that the steps were unable to produce a jet, and $f_s$ was set to zero. Finally, total $CO_2$ fluxes for each reach of the Swiss network were recalculated as

$$F_{CO_2}^* = F_{CO_2}\,(1+r) \tag{14}$$

which properly accounts for the localized $CO_2$ evasion occurring in correspondence with local steps. Note that, according to the assumptions introduced, the value of $r$ depends on the slope $i$ and the discharge $Q$ of a given reach.

## Data availability

The data that support the findings of this study are openly available in Botter et al. 2022[61] at http://researchdata.cab.unipd.it/id/eprint/619, reference number 619.

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

## Acknowledgements
This research was supported by the European Community's Horizon 2020 Excellent Science Programme (grant no. H2020-EU.1.1.-770999).

## Author contributions
G.B. and N.D. conceived and designed the study. G.B., N.D., and A.C. performed the experiments. A.C., G.B., P.P., and N.D. analyzed the data and discussed the results. G.B. wrote the main text, all authors wrote the Supplementary Information and reviewed the paper. A.C., N.D., and P.P. prepared the figures.

## Competing interests
The authors declare no competing interests.
