## [Peer Review File · Nature Communications]

Steps dominate gas evasion from a mountain headwater streamReviewer #1 (Remarks to the Author):

This manuscript addresses how variation in stream longitudinal profile within a reach can influence rates of gas exchange. Current models for gas exchange use large scale properties of a stream reach to predict k: slope, velocity, depth etc. and models with these terms are not very useful for accurately predicting k for any one stream. The authors here show that the variation of slope within any segment say one with steps or not, can control rates of gas exchange because steps are hotspots of such exchange. I think the next step, so to speak, in stream gas exchange research is to include local variation in morphology and hydraulics and the authors are showing us what needs to be done, great.

General comments:

1. It is hard to link the work in this paper to its actual implications for gas exchange in a stream reach. I think this point is because there are no estimates of K_{600} in this paper! There are derived values showing the fraction of gas coming from steps, but not the actual gas exchange rates themselves. It is difficult for me to link these derived values with how we might use this paper to design better models for predicting gas exchange. It would be beneficial to compare the authors' findings for gas exchange here with models that do not include steps. At the very least provide K_{600} data so data from this stream can be part of multi-stream models.

2. This paper uses variation in CO_2 across steps to estimate gas exchange. But CO_2 is a small part of a possibly much larger pool of dissolved organic carbon. How small depends on the alkalinity of the stream, and these data are not part of the paper. One feature of highly buffered streams is that CO_2 coming and going into the water will affect the pH and distribution of bicarbonate. How much does one need to take into account the rest of DIC pool when calculating the amount across a step. It may be small, but the authors may wish to do some chemical modeling here that includes the role of bicarbonate changing as CO_2 is lost or gained over steps.

3. The raison d'être of knowing gas exchange in the first place is to calculate gas fluxes. There are much CO_2 data in the paper, but no estimates of the fluxes.

4. It seems that the role of steps will be a function of the spatial scale of gas transport vs the distances between steps. If there are big steps at a scale that exceed gas turnover, then I expect these steps to matter a lot. But if there are lots of small steps then I would expect gas concentrations to be right near equilibrium

5. C_a is atmospheric CO_2 concentration but the authors never stated how they measured C_a and how variation in C_a may affect their conclusion. If they assumed 400 ppm or something like that, what if the CO_2 is higher due to terrestrial respiration? What is C_a at their site and how will its variation matter to estimates of gas exchange?

Specific comments

16,. I think this sentence overstates the implications of the paper. This is a study of one stream at one time and with no presented estimates of gas exchange.

24 high gas exchange in head water streams does not really control the flux of gas per se. It is the inputs as stated on line 23. The gas exchange controls where the CO_2 is emitted: close to where it entered the stream or further downstream

32 replaces systems with streams

45. The authors may wish to refer to Whitmore et al. Ecosphere 2021 in this paragraph

60. I am a little unclear here. If one measure gas exchange using a tracer gas over a

reach of stream that includes several steps, then won't the resulting K estimate include the effects of steps? I understand that estimate may be biased if a reach with few or many steps is selected.

100 delete proper

103 we use the concept

eq 1. What is x'

128 similar to

129 `` for left quotes

152 state the finding and not where to look for the finding. Where to look can be parenthetical

174 delete found to be

191 ditto

203 remarkable is unclear here, state the fractional contribution

222, 224. pick and or or

242 steps does not vanish

244-250. I am having difficulty understanding these sentences.

259 excess mass?

260 Key point here. How much CO₂ does emit from steps? There are no CO₂ data in the paper.

270. I agree with this point. It could be stronger if the authors used predictive modes to assess k₆₀₀ and then compared that with actual k₆₀₀ measured across the steps.

277 not sure what is meant by excess mass.

292. The math is split between 2 sections of the paper making it more difficult for me to follow

301 use an inline equation here to make this point

310 length

335 show these CO₂ data

368. The film gets mentioned in the methods but should be stated early as part of the overall study design. Overall the study design emerges as I read the methods. The design should come first.

415. Give these details in the methods

420. Won't this film change roughness?

444 This ____?

Fig 2. Statistical inference shows up in the figure legend, but there was nothing in the

methods stating these methods or really any probabilistic modeling or uncertainty at all. What is a police strobe? A color?

Reviewer #2 (Remarks to the Author):

Botter et al. 2022 manuscript submitted to Nature Communications

This study quantitatively describes the loss of CO₂ due to step pool morphology in a high gradient stream. The authors present an important advancement in better understanding greenhouse gas loss through evasion at a finer spatial scale that should be useful for many seeking to pin down this elusive loss in the fluvial carbon cycle. The development of concentration damping as a metric is both clever and relatively intuitive once the authors have walked through it, and it could be widely applied in other works. The manuscript is generally well written and novel in both findings and approach.

Key results:

The authors found that step pool morphology in a small stream leads to significant outgassing of CO₂ over what would be expected (for the same distance) from simple turbulence resulting from steep slope and rough streambeds. The amount of this influence seems to depend on step height and is cumulative over many steps throughout a reach. Because CO₂ 1) emissions can be highly heterogeneous over short spaces and 2) step pools may allow large percentages of CO₂ produced at local hotspots to escape quite quickly, the influence of step pools on stream contributions to larger-scale estimates of CO₂ emissions may be underestimated when these features are present in the network.

Validity and Significance:

The authors present significant and interesting conclusions. I believe the author's interpretation of the data as it pertains to their small study stream is quite sound and well-argued. I agree with their assertion that, given the commonality of step pool morphology in many fluvial networks, this work sets up important considerations of what researchers interested in stream outgassing may be neglecting or underestimating when working in other regions and in larger streams and rivers; however, I am skeptical of a few of the very broad attempts to claim their relationship will certainly apply at scale. Not only is the study stream and reach quite small, but the observed and modeled discharge ranged only from about 0.25 - <4 L/s. I do not think this flaw should alone prohibit publication, but it may serve the authors to qualify how far down the network they apply their findings.

Data and methodology:

The approach to answering the question, "how much do step pools influence stream outgassing?" is creative and generally thorough. It's curious that "step-and-pools" opens the abstract and step pools are mentioned often, but that the framing later shifts to steps and segments? It is not clear if the authors considered wherever the jet entered downstream to be the pool or if they are actually referring to geomorphic pools, as is in deeper, wider channel sections where travel time is typically slower and turbulence can be vastly reduced. Perhaps this latter type of pool was not present in the study reaches, but since they are often thought of with steps and cascades, and are ecologically and biogeochemically relevant, it may be helpful to mention any counters or amplifications of "dampening" that the authors think could occur when finding such pools associated with steps.

From the supplement, I would like to know how many paired CO₂ measurements were omitted from analysis because the upstream-downstream difference was not great enough? (lines 150-152) Depending on the number of omissions, it would also be helpful to know when and where these small differences occurred. If there are instances where the step does not cause a great difference, would that not be an important caveat

to the overall conclusions? It would be nice, from a fuller hydrobiogeochemical perspective, to see longer or more CO₂ time series over more varied conditions, e.g., wider range of discharge. I also wonder if there is a succinct way to incorporate real CO₂ data into the main text, instead of the hypothetical plots of Fig.1 (or if those are real data, make that clearer; see line note about figure 1).

Suggested improvements:

I suggest the authors better identify how their data from a single site can be applied to address broad knowledge gaps in river outgassing. Yes, steps are common in many catchments, but this raises a few questions that aren't addressed. Can you give, for example, even a crude density estimate for this watershed, for reference, instead of citing "the World"? How would you compare wider step pools (or with multiple jets) to the single jets of this project? Even mountain headwaters can have steps many times the width of this channel, i.e., not only are results based on a single small stream but a particularly small channel (understandable to make the piping/plastic cover feasible). Steps can occur in both higher and lower gradient streams, how might position and number relative to slope matter for % mass lost? See line comments for further things to consider relating to .

My expertise:

I am a biogeochemist focused on carbon dynamics in low order streams. I am not an expert in the physics of hydrological processes, though I appreciate, for reasons enumerated in this manuscript, their importance in biogeochemical cycles of fluvial systems.

Line comments:

30: Possible conflation/misleading description of pools in this study -- wording implies larger hollowed out pool bedform but later photos and descriptions do not maintain this

31: Again, step pool geomorphology describes a lot of incredibly diverse channel structures when considering all these regions and when trying to apply to larger networks

124: What is the difference between steps and cascades? size? If you are only dealing with steps here, suggest not mixing in other words, or if cascades signifies something unique be clearer what that is

145: Suggest not putting the opposite conditions in separated parentheses, as it makes the sentence clunky; something like "Likewise, if $r > 1$ then the steps proved a larger contribution to the outgassing, or if $r < 1$ a smaller outgassing, as compared to the contribution generated by the continuous turbulent stretches." is easier for the reader

166: This is a nice way to put this, very concrete for the reader

174: Makes sense, though it would still be good to see this proved across a wider range of Q; other changes that the channel and water body undergo, particularly at very high flows, would be interesting to learn about or for the authors to acknowledge

204: It would be beneficial to have a deeper discussion of why reconstructing r from discharge to demonstrate a temporal difference is useful; it seems you are only showing a modeled hydrologic difference but calling it hydroclimatic. I am missing where the seasonal element comes in separate from Q, and seasonal changes in CO₂ can be great, as the authors acknowledge elsewhere

247: This sentence is doing a lot of heavy lifting and the meaning gets a bit muddled, suggest rephrasing the first half especially.

250: This sentence seems a little logically counterproductive while somewhat

illustrating the very limitations of the conclusions-- put another way it could read "suitable to be applied to a variety of streams as long as you are applying to similar streams," so in fact not suitable to extrapolation at all? Perhaps be more clear about what you mean by scaling vs extrapolating. For example, could you scale to any larger stream with step pools? Or to a different underlying geophysical makeup as long as flow disruptions/falling waters are present? How similar does the morphology need to be?

Fig. 1: Including y-axis label of CO₂ concentration instead of relying on "Cu/d" will help orient the reader at this stage, especially if they have not yet seen the equation; I suggest putting the lines on one plot, as in the supplement, and keep the arrows but from the lines to the location in the photograph; as is it's hard to tell that the Cd line is lower than the Cu; make sure you are being clear about the differences between "reference" and "representative" reaches here and throughout (as well as where they go on the map), it was easy to mix these up. In general this figure gives a lot of helpful information, but is quite a bit to take in all at once, with such a lengthy caption. Perhaps consider splitting into 1a + 1c-e and 1b as a separate diagram, maybe closer to the equations.

Fig. 2: It may help to label the plots "step stream" (or similar) and "reference" more boldly in the figure rather than relying on the caption, I think it would help readers orient to the data more quickly since the axes are different. See note for fig. 3 regarding color choices as well.

Fig. 3: Use of the reference dashed line is very helpful. I think you could just call your "blue petrolatum" points "blue" and the "ibis-shaded" region orange, or light orange. Continued use of blue and orange to no longer signify upstream/downstream for both points and shading requires the reader to readjust--perhaps consider different contrasting colors?

Reviewer #3 (Remarks to the Author):

Overview and major comments:

The authors present an elegant and well-conceived experimental approach with robust results to estimate the impact of steps in stream systems on the exchange of gases between the atmosphere and the stream itself. The results suggest a critical gap in our ability to properly estimate the emission and aeration of gasses in small mountainous streams without including specific parameterization of step, thereby leading to potentially underestimates of emissions of CO₂. This is really interesting! I have not seen a manuscript that so clearly described the methods, caveats, and concerns related to each step of the experimental setup, as well as the issues with in-situ CO₂ sensors. I really appreciate the effort presented here and think that it is potentially a disservice to leave much of this work in the Supporting Information. I will return to this point. The promotion of the 'dampening' term here is nicely done and the core message is that perhaps the development of an approach to estimate 'r' or the dominance ratio, as a truly more integrative approach to scale the gas transfer rates across dynamic small stream systems should be pursued and explored. In fact, that is what I was looking for by the end of the manuscript to make this manuscript something that might be most appropriate for Nature journals.

As it reads now, I believe that this manuscript will be difficult for many of the Nature Communications audiences to understand, as it is presented in a very technical format. It is understandable that the authors must describe the integration of hydrology and chemistry through the equations presented, and that there are very key parameters that are being assessed in the context of exchange. However, the reader is currently lost at many points regarding at continuation of why we need to care about this. As alluded to above, by the end of the paper I was looking for some assessment of the impact that these findings could show. For example, could you take the Horgby et al approach for

small mountainous streams and run a sensitivity analysis across a series of dominance ratios in the context of a known distribution in the literature then scale up to what this would potentially mean for their estimates of stream emissions? This is the type of analysis that would make it a stronger contribution to this journal.

Without these larger-scale applications of the results, the manuscript belongs in a journal where all of the really important information that is locked away in the SI can be highlighted and discussed. The subject matter is cross-disciplinary between hydrology, biogeochemistry, stream ecology, and carbon cycle science, and has the potential to contribute a meaningful advancement that may increase our confidence in stream CO₂ emission estimates. Just the simple example in the SI section 1.6, if brought forward into the main text, would make this manuscript much more approachable to a wider audience beyond physical hydrology. Furthermore, if this paper is going to focus on CO₂, there are many opportunities to discuss the implications for CO₂, and the role that gaining and losing streams, biofilms, soil inputs, etc. may play in understanding these emission controls. Please consider this.

Minor Concerns:

Consider using language throughout that makes the reference and step reach more clearly separated.

It was slightly unclear how the damping factor and the flow were related with so few measurements and how these were extrapolated to other time periods in the study.

68: careful as we need to see the in-prep paper for this cited statement if it is the only one available.

88: While we recognize that the morphological complexity of rivers might not fit perceptual models of the type proposed here, we conceptualize high-gradient stream networks as heterogeneous sequences of two types of elements: steps and segments. Why might the morphological complexity of rivers not fit perceptual models of the type proposed here? This statement is confusing.

134 and other locations: be more specific regarding the location you are referring to in the supporting information.

Line 155-156: Please provide additional clarity when describing each of the percentages in this case related to mass. Corresponding to approximately 10% - 25% mass removed. What does this mean? Why does 0.32 correspond to 25%?

172-173: the finding regarding the linear relationship between F_s and step height is really interesting and could be very complicated to unravel, but this should be explored more if possible - even present a hypothesis here.

255: How will the sources of matter varying spatially affect these results?

269: How would these factors be incorporated into scaling laws?

277: Excess mass? Why does the percentage of excess mass go from a range to a single value?

284: How can these be taken into account? How can this be scaled up? What are the actual implications for large-scale studies?

348: What was the representative segment used for? I see below - describe this here.

Generally - can you provide more clarity on how the dominance ratio is or could be calculated over large reaches?

Figures and Tables:

Figure 1 – can't see the inset graphs and they don't have any units to even know if the upstream and downstream concentrations are the same.

Figure 2 – what is a police probe circle color? Perhaps you don't need to say the actual colors as you have them in the figure key.

Figure 3 – again the names of the label colors (blue petroleum) are not needed and are distracting - or just call it a simple name. A and C sort of say the same thing right - consider if you need both and are these actually referenced in the paper? They are only briefly discussed in lines 204-205, as such you might consider their importance to the audience.

Support Information:

Please consider a map so we know where you are, and what this stream may look like within the southern Alps. Photos do not show the slope very well.

Please consider much of this to be worthy methods and material for discussion in a longer format journal.

Response to the Referees

We would like to thank the reviewers for handling our work, and for giving us the opportunity to revise the Ms. In what follows, a point-by-point response to all the comments is presented. In this rebuttal, text taken from the decision letter is presented as blue indented text. Each comment is followed by a detailed response, with a list of actions implemented. Where specified, the line numbers refer to the manuscript version with tracked changes. The large majority of the comments received were properly complied with. This hopefully helped us to improve the paper and make it suitable to the audience of Nature Communications.

Reviewer #1 (Remarks to the Author)

We thank the referee for the insightful comments, and for his/her assessment of the paper. All the comments formulated have been accommodated by adding new information to the paper, changing the text, or clarifying the meaning of some unclear statements.

This manuscript addresses how variation in stream longitudinal profile within a reach can influence rates of gas exchange. Current models for gas exchange use large scale properties of a stream reach to predict k : slope, velocity, depth etc. and models with these terms are not very useful for accurately predicting k for any one stream. The authors here show that the variation of slope within any segment say one with steps or not, can control rates of gas exchange because steps are hotspots of such exchange. I think the next step, so to speak, in stream gas exchange research is to include local variation in morphology and hydraulics and the authors are showing us what needs to be done, great.

Thank you for recognizing the important contribution of our work, and for the positive evaluation of the paper.

General comments

1. It is hard to link the work in this paper to its actual implications for gas exchange in a stream reach. I think this point is because there are no estimates of K_{600} in this paper! There are derived values showing the fraction of gas coming from steps, but not the actual gas exchange rates themselves. It is difficult for me to link these derived values with how we might use this paper to design better models for predicting gas exchange. It would be beneficial to compare the authors' findings for gas exchange here

with models that do not include steps. At the very least provide K600 data so data from this stream can be part of multi-stream models.

Thanks for the comment. k_{600} values were added to the Ms, when possible (line 196 and Supplementary Table S4). However, we would like to stress that k_{600} values are meaningful only for the segments and not for the steps, as explained in lines 61-68 of the previous version of the Ms. This is due to the fact that the value of mass transfer rate which has to be associated to a step is largely dependent on the volume of water over which k is calculated. In the numerical simulation reported in Supplementary Text 1.1 and Supplementary Figure S1, one can see how spatially variable could be the gas transfer rate in correspondence of a step: right downstream of the jet k can be 10 to 100 times larger than the mass transfer rate observed in the downstream pool, or in the ramp. Thus, if we average k over a very small volume in correspondence of the falling jet, we could get very large values - while if we take the average value of k in the entire step-pool system, the mass transfer rate would be much lower. Provided that in real-world situation the spatial distribution of k is not known, the volume of water which has to be considered to be representative of the step is not known a priori. Therefore, owing to this scale-dependence of k , the value of the mass transfer rate of a given step is quite meaningless. At most, we could provide values of k for reaches containing some steps, but this would be dependent on the number of steps and their geometric features. Moreover, in our case, we have measured the steps and the segments separately, so we do not have directly measured the outgassing of composite reaches. Rather, we provided the basic elements to estimate the outgassing of composite reaches. Incidentally, we would like to stress that the damping factors provided in this study could be directly compared with previous studies by calculating f from available k , h and τ data ($f = (k\tau)/h$) and this comparison is now shown in Supplementary Figure S11. All these arguments were better clarified in the revised text (lines 71-84).

2. This paper uses variation in CO₂ across steps to estimate gas exchange. But CO₂ is a small part of a possibly much larger pool of dissolved organic carbon. How small depends on the alkalinity of the stream, and these data are not part of the paper. One feature of highly buffered streams is that CO₂ coming and going into the water will affect the pH and distribution of bicarbonate. How much does one need to take into account the rest of DIC pool when calculating the amount across a step. It may be small, but the authors may wish to do some chemical modeling here that includes the role of bicarbonate changing as CO₂ is lost or gained over steps.

Thank you for pointing this out. The pH of the stream water has been measured through a YSI sensor during our field campaign, with an accuracy of 0.1. During the experiments, we have measured the pH of the water flow upstream and downstream of each segment and step, and we have not detected sizable differences in pH with our sensor. During the field campaign, the pH was always

in the range between 7.6 and 8.1. In this range of alkalinity, potential changes of CO_2 concentration due to small changes in water pH are known to be very limited [1]. Therefore, we have disregarded the effect of pH spatial variations within the segment or the steps (below the detection limit of our instrument) on the estimated damping factors. This was specified in the revised text (lines 548 and 628-631).

3. The *raison d'être* of knowing gas exchange in the first place is to calculate gas fluxes. There are much CO_2 data in the paper, but no estimates of the fluxes.

Thank you for the comment. The referee is certainly right when he/she says that gas fluxes are important. The observed values of the fluxes during our field campaign were thus included in the revised text (lines 186-191, Supplementary Table S4 and S6). However, it is important to stress that the fluxes are strongly dependent on the underlying water CO_2 concentration, which in turn depends on a number of important biological and climatic agents that fluctuate in time at all the relevant timescales (hourly, daily, weekly, seasonally). Instead, the damping factors (which are the core of our paper) mirrors physical processes that change through time much more slowly - only when the discharge varies. Thus, while the observed damping factors can be seen as representative of the gas evasion potential of a given stream element for a given discharge level, the corresponding observed flux could change hour by hour, or day by day depending on other external biological and climatic factors impacting the stream CO_2 concentration. At any rate, we have also proposed a regional-scale application, in which we have used the method developed in the paper to estimate the CO_2 fluxes across water air interfaces of Swiss mountain streams (lines 455-471 and Fig. 5).

4. It seems that the role of steps will be a function of the spatial scale of gas transport vs the distances between steps.

The referee is correct. The spatial frequency of the steps is an important driver of gas exchange in step-pools rivers, and one of the reasons that explain the dominance of the steps over the segments is their relatively high spatial frequency, as already noted in the text (lines 332-335).

4. If there are big steps at a scale that exceed gas turnover, then I expect these steps to matter a lot. But if there are lots of small steps then I would expect gas concentrations to be right near equilibrium

The damping factors and the dominance ratios do not depend on the underlying water CO_2 concentration. As per the role of big vs. small steps, in the paper we were actually able to quantify the relative role of steps with different heights, and we found out that small steps are more important than large steps, owing to their higher spatial frequency. In particular, more than 60% of the gas evasion

form the steps was found to be associated to steps with an height smaller than 35 *cm* (line 254-256 and Supplementary Table S8). As per the value of water CO_2 concentration downstream of a step (or a segment), this could be either near the equilibrium with the atmosphere or not, depending on the spatial patterns of gas evasion and CO_2 production/input. In our settings CO_2 concentrations were typically above (or close to) 1000 ppm. Thus, the equilibrium with the atmosphere could not be reached along the focus reach. The typical values of water CO_2 concentrations in the focus reach were added to the text (line 187).

5. C_a is atmospheric CO2 concentration but the authors never stated how they measured C_a and how variation in C_a may affect their conclusion. If they assumed 400 ppm or something like that, what if the CO2 is higher due to terrestrial respiration? What is C_a at their site and how will its variation matter to estimates of gas exchange?

Atmospheric CO_2 concentrations were measured with the same sensor used for water CO_2 concentrations after each saturation curve observed in water. Observed atmospheric concentrations were nearly constant during the study period, in the range between 390 and 412 ppm. These values were in substantial agreement with the nearest stations of the World Data Center for Greenhouse gases (Monte Cimone, Sonnblick Observatory, Zugspitze), where carbon dioxide concentrations in the atmosphere are continuously recorded using high-precision instruments. A sensitivity analysis also revealed a negligible impact of the observed variations of atmospheric CO_2 concentrations on the damping factors (e.g. 20 ppm of increase/decrease in C_a implies an increase/decrease of less than 4% in the damping factor in all the cases analyzed). This information has been added to the revised text (lines 632-638).

Specific comments

16,. I think this sentence overstates the implications of the paper. This is a study of one stream at one time and with no presented estimates of gas exchange.

This is a study of one stream, observed several times during the summer of 2021, but we believe the take home message is quite general. The limitations of the study implied by the practical difficulties in extending the field measurements to other sites were actually discussed in the text (lines 306-308). Estimates of gas exchange during the field experiment were also added to the text, as per the previous request of the referee. Moreover, in the revised text we have elaborated more on the potential reasons for which current estimates of large-scale gas exchange could be biased (lines 450-481) to justify the final statement of the abstract. In particular, a novel estimate of CO_2 fluxes from Swiss mountain streams has been added to the Ms. (lines 455-471).

24 high gas exchange in head water streams does not really control the flux of gas per se. It is the inputs as stated on line 23. The gas exchange controls where the CO₂ is emitted: close to where it entered the stream or further downstream

Thanks for the comment. The flux of gas from headwater streams in a given reach is high if the input is high and the emission capacity of the reach is high (otherwise the CO₂ will be simply routed downstream). We did not mean to state that “gas exchange in head water streams does control the flux of gas per se”. The sentence was rephrased to better highlight this fact (line 23-25).

32 replaces systems with streams

Done, thank you.

45. The authors may wish to refer to Whitmore et al. Ecosphere 2021 in this paragraph

Done, thank you.

60. I am a little unclear here. If one measure gas exchange using a tracer gas over a reach of stream that includes several steps, then won't the resulting K estimate include the effects of steps? I understand that estimate may be biased if a reach with few or many steps is selected.

The referee is correct. The reasoning proposed above is a key implication of our findings, and thus it has been briefly discussed in the revised Ms. at lines 473-481. However, we are not able to understand the link with line 60. Maybe the referee is referring to lines 61-67, where however the point we are trying to make refers to the fact that the value of spatially averaged K values around a single step are essentially dependent on the unknown averaging volume. These lines have been rewritten in the revised introduction (lines 71-84).

100 delete proper

Done, thank you.

103 we use the concept

Agreed, thanks.

eq 1. What is x'

x' is the integration variable, representing an arbitrary position within a reach with length x. This has been clarified in the revised text (lines 510).

128 similar to

Agreed, thank you.

129 “ for left quotes

Quotes have been removed. Thank you.

152 state the finding and not where to look for the finding. Where to look can be parenthetical

The statement has been rephrased. Thanks.

174 delete found to be

Fixed, thank you.

191 ditto

Fixed, thanks.

203 remarkable is unclear here, state the fractional contribution

The statement has been rephrased.

222, 224. pick and or or

Done, thanks.

242 steps does not vanish

Corrected, thank you.

244-250. I am having difficulty understanding these sentences.

The sentences quoted above have been removed from the text. Thank you.

259 excess mass?

The excess mass is the mass that exceed the mass transported by the river when the system is in equilibrium with the atmosphere (i.e., $Q C_a$). The excess mass is now defined in line 135-136.

260 Key point here. How much CO₂ does emit from steps? There are no CO₂ data in the paper.

CO₂ data are shown in Figures S8 and S9. CO₂ fluxes has been added to the Ms (lines 188-191). Thank you.

270. I agree with this point. It could be stronger if the authors used predictive modes to assess k600 and then compared that with actual k600

measured across the steps.

As discussed above, k_{600} values for individual steps can not be calculated, as they depend on the size of the water volume that is used for their calculation. If a small volume of 1 cm^3 right below the falling jet is selected, the corresponding k will be very large. If k is averaged over the pool and the ramp, the resulting k will be much lower. Thus the value of k is entirely dependent on the choice of the reference volume which is however not known - and not identifiable a priori. This is discussed in lines 71-84.

277 not sure what is meant by excess mass.

The excess mass is now defined in line 135 and 515.

292. The math is split between 2 sections of the paper making it more difficult for me to follow

All the math has been moved to the methods, thank you.

301 use an inline equation here to make this point

We prefer to keep the text as it is, as this point is not directly relevant to the paper, which is focused on the damping factor and not on k .

310 length

Fixed, thank you.

335 show these CO₂ data

All the relevant CO₂ data are shown in the SI and Figure 1.

368. The film gets mentioned in the methods but should be stated early as part of the overall study design. Overall the study design emerges as I read the methods. The design should come first.

A new subsection about the study design has been created in the methods. Thank you.

415. Give these details in the methods

Methods have been expanded. We have moved some text from the main Ms. and the SI to the methods and we have created a new subsection in the Methods about the study design, as per the suggestion of the referee. Moreover, we have added more details also to this subsection (see below).

420. Won't this film change roughness?

We have checked that this is not really the case by evaluating the travel times under covered and natural conditions (Supplementary Fig. S5). the difference in the travel times is quite limited for the whole ranges of discharges identified, suggesting that the impact of the film on the roughness is quite limited. This is mainly related to the fact that debris and stones were relocated over the plastic film before let the stream was let to flow back over the film. Some of this information has been added to the revised methods (lines 594-600).

444 This ----?

Apologizes, but we didn't understand the above comment and we didn't know how to take actions here.

Fig 2. Statistical inference shows up in the figure legend, but there was nothing in the methods stating these methods or really any probabilistic modeling or uncertainty at all. What is a police strobe? A color?

Color names have been changed. More details about the regression have been also added to the text. As the least-squared linear regression is a standard method in scientific studies, we have added only a few remarks in the Figure caption.

Reviewer #2 (Remarks to the Author)

We thank the referee for the insightful comments, and for his/her assessment of the paper. All the comments formulated have been accommodated and they have been instrumental to improve the quality of the paper.

This study quantitatively describes the loss of CO₂ due to step pool morphology in a high gradient stream. The authors present an important advancement in better understanding greenhouse gas loss through evasion at a finer spatial scale that should be useful for many seeking to pin down this elusive loss in the fluvial carbon cycle. The development of concentration damping as a metric is both clever and relatively intuitive once the authors have walked through it, and it could be widely applied in other works. The manuscript is generally well written and novel in both findings and approach.

We really appreciated the positive assessment of the novelty of the paper provided by the Referee.

Key results: The authors found that step pool morphology in a small stream leads to significant outgassing of CO₂ over what would be expected (for the same distance) from simple turbulence resulting from steep slope and

rough streambeds. The amount of this influence seems to depend on step height and is cumulative over many steps throughout a reach. Because CO₂ 1) emissions can be highly heterogeneous over short spaces and 2) step pools may allow large percentages of CO₂ produced at local hotspots to escape quite quickly, the influence of step pools on stream contributions to larger-scale estimates of CO₂ emissions may be underestimated when these features are present in the network.

We thank the referee for the nice summary of the main points of our Ms.

Validity and Significance: The authors present significant and interesting conclusions. I believe the author's interpretation of the data as it pertains to their small study stream is quite sound and well-argued. I agree with their assertion that, given the commonality of step pool morphology in many fluvial networks, this work sets up important considerations of what researchers interested in stream outgassing may be neglecting or underestimating when working in other regions and in larger streams and rivers;

Thank you once again for the positive evaluation of our paper.

however, I am skeptical of a few of the very broad attempts to claim their relationship will certainly apply at scale. Not only is the study stream and reach quite small, but the observed and modeled discharge ranged only from about 0.25– < 4L/s. I do not think this flaw should alone prohibit publication, but it may serve the authors to qualify how far down the network they apply their findings.

We understand the concern of the referee. The range of discharges used for measuring f_c is very low in fact - as diverting a bigger rivers would be practically infeasible. However, in the revised text we have provided evidence of the fact that our f_c estimates are within the range of previously observed values as derived from tracer experiments under a much wider range of discharges (Figure S11). As per the observed values of f_s , our experimental points do include one step of the main Valfredda river, which experienced a discharge of about 109 l/s. We feel we have to thank the referee for rising this important point - in fact, we noticed that this information was not included in the previous version of the paper (and was added to the text, see lines 229-231). The point in figure 3a of the revised paper (former Fig 2a) that corresponds to the step with a discharge of more than 100 l/s is perfectly aligned with the other steps, suggesting that f_s indeed does not depend on Q - which is reflected by the lack of correlation between f_s and Q (Table S6). We agree on the general concern that extrapolating our results to larger Q values could be potentially problematic, but we are also confident that our main claims should not be case-specific. In the revised version of the paper, we have elaborated more on the potential limits of our analysis, and the related implications (line 342-350)

Data and methodology: The approach to answering the question, “how much do step pools influence stream outgassing?” is creative and generally thorough.

Let us thank once again the referee for her/his interest in the research question investigated in the paper.

It’s curious that “step-and-pools” opens the abstract and step pools are mentioned often, but that the framing later shifts to steps and segments? It is not clear if the authors considered wherever the jet entered downstream to be the pool or if they are actually referring to geomorphic pools, as is in deeper, wider channel sections where travel time is typically slower and turbulence can be vastly reduced. Perhaps this latter type of pool was not present in the study reaches, but since they are often thought of with steps and cascades, and are ecologically and biogeochemically relevant, it may be helpful to mention any counters or amplifications of “dampening” that the authors think could occur when finding such pools associated with steps.

Thanks for the comment, which gave us the opportunity to clarify the matter. The energy dissipation and the outgassing process in corresponding of a plunging jet is highly local, as shown by Figure S1 - which was added to the paper during this round of revisions. Therefore, the presence of the pool, where the water is essentially standing, per se does not significantly impact the outgassing process - although being very relevant for the underlying biogeochemical processes and sediment transport. In fact we have observed experimentally that the concentration of a non-reactive tracer within the pool and below the falling jet is nearly uniform. For this reason, the pools - which in some cases were detected downstream of the steps - in our paper were considered as part of the segments. Notice that the above choice does not impact the value of the damping factors, but only the corresponding travel times τ and the physical size of the stream elements considered in the paper, the “segments” and the “steps”. We believe that the choice to include the pools (if any) in the segments not only complies with the local nature of the outgassing process in the plunging jet but also complies with the practical difficulty of recreating pools in artificially generated steps. These arguments were better specified in the revised paper (line 115-119). Moreover, to avoid any confusion we have replaced the term “step and pools” with “steps” throughout the paper, with only a few motivated exceptions in the introduction.

From the supplement, I would like to know how many paired CO₂ measurements were omitted from analysis because the upstream-downstream difference was not great enough? (lines 150-152) Depending on the number of omissions, it would also be helpful to know when and where these small differences occurred. If there are instances where the step does not cause a great difference, would that not be an important caveat to the overall conclusions? It would be nice, from a fuller hydrobiogeochemical perspec-

tive, to see longer or more CO₂ time series over more varied conditions, e.g., wider range of discharge. I also wonder if there is a succinct way to incorporate real CO₂ data into the main text, instead of the hypothetical plots of Fig.1 (or if those are real data, make that clearer; see line note about figure 1).

All the relevant CO₂ timeseries were plotted and reproduced in the SI and in Figure 1. The number of paired measurements which were eliminated was quite limited. All the omissions corresponded to steps with limited height and relatively low CO₂ concentrations, for which the corresponding concentration drops ΔC were too low. This information has been added to the revised text (see Supplementary Text 1.5.4). Longer timeseries would not be particularly helpful instead, as the sensor was moved continuously from place to place during any survey. As the referee guessed in the above comment, Figure 1 does contain real data, and this was specified in the revised text.

Suggested improvements: I suggest the authors better identify how their data from a single site can be applied to address broad knowledge gaps in river outgassing.

Thank you for the comment. We have provided more text to discuss the potential implications of our work. We have also added one example application in which we have discussed quantitatively the potential role of steps in large-scale applications (lines 455-471).

Yes, steps are common in many catchments, but this raises a few questions that aren't addressed. Can you give, for example, even a crude density estimate for this watershed, for reference, instead of citing "the World"? How would you compare wider step pools (or with multiple jets) to the single jets of this project? Even mountain headwaters can have steps many times the width of this channel, i.e., not only are results based on a single small stream but a particularly small channel (understandable to make the piping/plastic cover feasible). Steps can occur in both higher and lower gradient streams, how might position and number relative to slope matter for mass lost? See line comments for further things to consider relating to

In the revised discussion, We have tried to address more in depth the questions posed by the referee in the above comment. Specifically, we have added reference numbers about the frequency and height of steps in our set up, and in other contexts where these data were available (line 310-318).

My expertise: I am a biogeochemist focused on carbon dynamics in low order streams. I am not an expert in the physics of hydrological processes, though I appreciate, for reasons enumerated in this manuscript, their importance in biogeochemical cycles of fluvial systems.

We would like to thank the referee once again for her/his insightful comments, which definitely helped us to straighten our Ms.

Line comments

30: Possible conflation/misleading description of pools in this study – wording implies larger hollowed out pool bedform but later photos and descriptions do not maintain this

Several pools have been actually observed in our study reach, but their presence / absence is not relevant for the damping factors and the outgassing processes - which is instead a very local process, as discussed above. We have clarified the matter in the revised text (lines 115-119), and avoided the use of the term pools in the paper.

31: Again, step pool geomorphology describes a lot of incredibly diverse channel structures when considering all these regions and when trying to apply to larger networks

Of course, we agree with the referee. This has been clarified in the revised text (lines 29-39).

124: What is the difference between steps and cascades? size? If you are only dealing with steps here, suggest not mixing in other words, or if cascades signifies something unique be clearer what that is

we have removed "and cascades" (line 153)

145: Suggest not putting the opposite conditions in separated parentheses, as it makes the sentence clunky; something like "Likewise, if $r > 1$ then the steps proved a larger contribution to the outgassing, or if $r < 1$ a smaller outgassing, as compared to the contribution generated by the continuous turbulent stretches." is easier for the reader

Agreed, thank you (line 175-179).

166: This is a nice way to put this, very concrete for the reader

We thank the reviewer for his/her nice words. Thank you!

174: Makes sense, though it would still be good to see this proved across a wider range of Q; other changes that the channel and water body undergo, particularly at very high flows, would be interesting to learn about or for the authors to acknowledge

Thanks for the comment. See our previous responses on the same point. While

in our study f_s was observed only for values of Q in the range from 0.7 to 109 l/s, more experimental work would be needed to assess the independence of f_s on Q also at higher discharge levels. This has been specified in the revised discussion (lines 346-350). Moreover, in the revised text we have also formulated a research hypothesis on the theoretical reasons that underlie the independence of f_s on Q , as per the suggestion of the Referee 3 (lines 231-236).

204: It would be beneficial to have a deeper discussion of why reconstructing r from discharge to demonstrate a temporal difference is useful; it seems you are only showing a modeled hydrologic difference but calling it hydroclimatic. I am missing where the seasonal element comes in separate from Q , and seasonal changes in CO2 can be great, as the authors acknowledge elsewhere

River flow regimes depend on a number of intertwined hydrologic and climatic factors, such as precipitation, temperature, partial water pressure deficit in the atmosphere, etc. This is mainly reflected in the frequency of runoff producing events, which is in fact the byproduct of soil moisture dynamics (including evapotranspiration). The very reason for which we adopted the term “hydroclimatic” in this context is to summarize all the relevant hydrologic and climatic processes that leave a footprint in the the probability density function of the discharges (in line with the hydrological literature, see e.g. Botter et al., 2013²). This hydroclimatic variations are in general taking place both during a given season, and across seasons. In this specific case, however, intra-seasonal variations dominate - owing to the limited duration of the study period. All these arguments have been clarified in the revised text, in which we have removed the term “hydroclimatic” to avoid confusion of any kind.

As per the significance of the temporal dynamics of r simulated with the hydrological model, we do believe it is an important point of the paper. We have observed the value of r in our study reach only 4 times during one season. Consequently it was important to assess whether these spot measurements were representative of the long term behaviour of the system, or not. The analysis shown in Figure 4 quantifies how frequently different discharge levels (and thus different values of r) could be observed in the study river, and to what extent our measurements are representative of the long-term functioning of the study stream. The analysis presented in the paper shows that: i) the four experimental points of our study do cover almost all the range of observed discharges during the reference season; ii) the values of r obtained from our analysis are not biased (e.g. because we simply cherry-picked unusual hydrologic conditions during which r was particularly high). The mean seasonal values of r , calculated via Figure 4 properly weighting the relative frequency of each discharge range, are thus way more representative of the outgassing potential of the study reach than the simple algebraic average of the 4 measured values of r . This proves true in general, regardless of the temporal variations of C , which are certainly important for the calculation of the fluxes but not for the calculation

of the outgassing potential of the steps, which is the goal of this study. All these arguments has been clarified in the revised text (line 267-276).

247: This sentence is doing a lot of heavy lifting and the meaning gets a bit muddled, suggest rephrasing the first half especially.

We have rephrased this sentence, thank you (lines 360-367).

250: This sentence seems a little logically counterproductive while somewhat illustrating the very limitations of the conclusions– put another way it could read “suitable to be applied to a variety of streams as long as you are applying to similar streams,” so in fact not suitable to extrapolation at all? Perhaps be more clear about what you mean by scaling vs extrapolating. For example, could you scale to any larger stream with step pools? Or to a different underlying geophysical makeup as long as flow disruptions/falling waters are present? How similar does the morphology need to be?

The quoted sentence has been removed from the paper, and a new paragraph has been added to the Discussion to explain how the results presented in the paper could be scaled or extrapolated to other settings (lines 332-370). A real-world example of how our results could be used to make predictions of CO_2 fluxes at the regional level has been presented (lines 455-471). While extrapolations has to be taken with caution, we believe the example clarifies to the Nature audience the importance of our findings (see comment of Referee 3 on this point).

Fig. 1: Including y-axis label of CO_2 concentration instead of relying on “Cu/d” will help orient the reader at this stage, especially if they have not yet seen the equation; I suggest putting the lines on one plot, as in the supplement, and keep the arrows but from the lines to the location in the photograph; as is it’s hard to tell that the Cd line is lower than the Cu; make sure you are being clear about the differences between “reference” and “representative” reaches here and throughout (as well as where they go on the map), it was easy to mix these up. In general this figure gives a lot of helpful information, but is quite a bit to take in all at once, with such a lengthy caption. Perhaps consider splitting into 1a + 1c-e and 1b as a separate diagram, maybe closer to the equations.

Figure 1 contains real concentration data, and this was specified in the revised text. The figure has been modified as per the suggestion of the Referee. In the revised version the concentrations are shown in a single plot, and the Figure was split into two parts. Thank you.

Fig. 2: It may help to label the plots “step stream” (or similar) and “reference” more boldly in the figure rather than relying on the caption, I think it would help readers orient to the data more quickly since the axes are different. See note for fig. 3 regarding color choices as well.

The Figure has been modified as per the suggestion of the Referee: colors have been changed in Fig 2 and 3, and new labels have been added.

Fig. 3: Use of the reference dashed line is very helpful. I think you could just call your “blue petrolatum” points “blue” and the “ibis-shaded” region orange, or light orange. Continued use of blue and orange to no longer signify upstream/downstream for both points and shading requires the reader to readjust—perhaps consider different contrasting colors?

The Figure has been revised and the colors (and color names) have been changed. Thank you.

Reviewer #3 (Remarks to the Author)

Overview and major comments

The authors present an elegant and well-conceived experimental approach with robust results to estimate the impact of steps in stream systems on the exchange of gases between the atmosphere and the stream itself. The results suggest a critical gap in our ability to properly estimate the emission and aeration of gasses in small mountainous streams without including specific parametrization of step, thereby leading to potentially underestimates of emissions of CO₂. This is really interesting! I have not seen a manuscript that so clearly described the methods, caveats, and concerns related to each step of the experimental setup, as well as the issues with in-situ CO₂ sensors. I really appreciate the effort presented here and think that it is potentially a disservice to leave much of this work in the Supporting Information. I will return to this point. The promotion of the ‘dampening’ term here is nicely done and the core message is that perhaps the development of an approach to estimate ‘r’ or the dominance ratio, as a truly more integrative approach to scale the gas transfer rates across dynamic small stream systems should be pursued and explored. In fact, that is what I was looking for by the end of the manuscript to make this manuscript something that might be most appropriate for Nature journals.

We thank the referee for the time dedicated to review the paper, for the set of constructive comments provided, and for his/her positive assessment of the Ms.

As it reads now, I believe that this manuscript will be difficult for many of the Nature Communications audiences to understand, as it is presented in a very technical format. It is understandable that the authors must describe the integration of hydrology and chemistry through the equations presented, and that there are very key parameters that are being assessed

in the context of exchange. However, the reader is currently lost at many points regarding at continuation of why we need to care about this. As alluded to above, by the end of the paper I was looking for some assessment of the impact that these findings could show. For example, could you take the Horgby et al approach for small mountainous streams and run a sensitivity analysis across a series of dominance ratios in the context of a known distribution in the literature then scale up to what this would potentially mean for their estimates of stream emissions? This is the type of analysis that would make it a stronger contribution to this journal.

Thanks for the comment. We have improved the paper undertaking two major actions. First, we have rewritten and expanded the discussion and we have removed some technicalities from the main text about the damping factor, so as to better emphasize the impact of our work to a broad audience. Second, we have added a regional scale application, as suggested by the referee, through which we have quantified gas emissions from Swiss mountain streams taking into account the steps. The results are very interesting, and we feel that adding this application made a stronger contribution to the Journal, as foreseen by the Referee.

Without these larger-scale applications of the results, the manuscript belongs in a journal where all of the really important information that is locked away in the SI can be highlighted and discussed. The subject matter is cross-disciplinary between hydrology, biogeochemistry, stream ecology, and carbon cycle science, and has the potential to contribute a meaningful advancement that may increase our confidence in stream CO₂ emission estimates.

See our previous response on the same point.

Just the simple example in the SI section 1.6, if brought forward into the main text, would make this manuscript much more approachable to a wider audience beyond physical hydrology.

The example was already discussed in the main text, and its implications have been further expanded in the revised text (lines 419-436). We thank the referee for the comment.

Furthermore, if this paper is going to focus on CO₂, there are many opportunities to discuss the implications for CO₂, and the role that gaining and losing streams, biofilms, soil inputs, etc. may play in understanding these emission controls. Please consider this.

We understand the point raised by the referee. In a sense, the paper focuses on the CO₂ as the experimental measurements were done using CO₂ and the regional scale application presented in the revised version of the paper focuses on CO₂. However, the very goal of the Ms. is represented by the physical

processes that control gas evasion in a broader sense, in particular the outgassing in correspondence of the plunging jet of a step. Thus we feel our paper is more about gas transport and gas exchange with the atmosphere in general, beyond CO_2 . While we recognize the importance of hydrologic and biologic controls on CO_2 concentrations alluded by the Referee, we think including a in-depth discussion of such a broad topic would be distracting for the reader, given the scope of the paper. Therefore, no action was undertaken in this case.

Minor Concerns

Consider using language throughout that makes the reference and step reach more clearly separated.

We have changed the terminology in the paper avoiding the use of the term “reference” outside the context of the “reference segment”. Moreover, we have eliminated the double use of the terms “representative” and “reference”, which was clearly confusing. Reaches A , B and A^* were termed as “target reaches” throughout the revised paper. The subdivision of a reach in segments and steps is presented in the text (lines 105-119) and in the revised Figure 2. Thus, we believe that in the revised Ms. any potential ambiguity should be resolved.

It was slightly unclear how the damping factor and the flow were related with so few measurements and how these were extrapolated to other time periods in the study.

We used four measurements in the segment and 19 measurements in the steps to calculate the damping factor as a function of Q (for detailed info, see Supplementary Text S1.5). While the behaviour could be potentially misrepresented in Figure 4a of the revised version of the paper, in all cases r was close to 1 or larger than 1. Therefore, while we recognize the limits of this procedure - which was related to the complexity of the field measurements - we believe that more complex patterns in the $r(Q)$ function should not change the main point of the paper. This is now specified in line 256-264.

68: careful as we need to see the in-prep paper for this cited statement if it is the only one available.

For the sake of clarity, we have included an additional figure in revised SI (Figure S1), in which we show the expected trend of k_{600} as a function of the length of the integration scale in an idealized step-and-pool whose hydrodynamics was solved numerically. In particular, the colored map of the turbulent energy dissipation rate at the free surface clearly shows the marked heterogeneity of the dissipative process responsible for the gas evasion at the water-air interface, which occurs mostly in correspondence of the plunging jet.

88: While we recognize that the morphological complexity of rivers might not fit perceptual models of the type proposed here, we conceptualize high-gradient stream networks as heterogeneous sequences of two types of elements: steps and segments. Why might the morphological complexity of rivers not fit perceptual models of the type proposed here? This statement is confusing.

We refer to the fact that in some cases the steps can not involve the entire cross section, and that the distinction among riffles and steps may be difficult to assess on an objective basis. Thus the conceptualization proposed here (reaches = segments + steps) might not perfectly describe all settings. As the above issue is already discussed at the beginning of the discussion, we have removed the statement from the text at this point. Thank you.

134 and other locations: be more specific regarding the location you are referring to in the supporting information.

Thank you for the comment, this has been fixed throughout the whole Ms.

Line 155-156: Please provide additional clarity when describing each of the percentages in this case related to mass. Corresponding to approximately 10% - 25% mass removed. What does this mean? Why does 0.32 correspond to 25

The reason that underlies this correspondence is that the fraction of mass removed can be calculated as $1 - e^{-f}$. This information has been added to the text (lines 513-516 and eq. (8)). Likewise, the calculations that underlie the percentages shown here have been added to the text to clarify the procedure (line 421-423 and 428). We feel we have to thank the referee for raising this issue, as the previous version of the paper was quite unclear with this respect.

172-173: the finding regarding the linear relationship between f_s and step height is really interesting and could be very complicated to unravel, but this should be explored more if possible - even present a hypothesis here.

Thank you for the interesting comment. As highlighted by the reviewer, the reason for the linear relationship between f_s and Δh is not easy to guess due to the complexity of the processes involved. Nevertheless, our data allowed us to formulate some hypotheses, which were better articulated in the revised version of the paper (lines 233-236). The lack of a clear relationship between the damping factor and the flow rate seems to imply that ε , which dissipates most of the jet power impacting the free surface ($\varepsilon \propto P = \gamma Q \Delta h$), does not control the gas evasion in the case of a plunging jet. Conversely, the linear dependence of f_s on the step height suggests that bubble-mediated transport - which is possibly driven by the total jet energy, in turn proportional to Δh - could be the main driver of gas exchange in the case of steps.

255: How will the sources of matter varying spatially affect these results?

A new statement has been added to the text to better address this issue, thank you (lines 375-379).

269: How would these factors be incorporated into scaling laws?

The topic has been discussed more in depth in the revised version of the discussion (lines 332-370). At this stage is not easy to be more detailed about this issue, as long as information about the geometry of the reaches used for the development of existing scaling laws are not available. All these arguments were discussed in the revised version of the paper.

277: Excess mass?

The term "excess mass" is now defined in lines 136 (main) and 515 (methods) of the revised Ms.

Why does the percentage of excess mass go from a range to a single value?

This is because in the case of steps the damping factor is no longer dependent on Q . This has been specified in the revised text (lines 425-428).

284: How can these be taken into account? How can this be scaled up? What are the actual implications for large-scale studies?

Thank you for the comment. We have completely rewritten the last part of the discussion and we have included information about how f_c , f_s and r can be extrapolated/scaled. We have also discussed the potential implications for large scale studies, with the aid of one upscaling example referred to the flux of CO_2 from Swiss mountain streams. The application shows that, if previous studies have disregarded the role of steps, the flux of CO_2 from Swiss mountain stream would be 3 times larger than that foreseen by Horgby and co-authors.

348: What was the representative segment used for? I see below – describe this here.

This information has been added to the text at this point (line 570-573). Thank you.

Generally – can you provide more clarity on how the dominance ratio is or could be calculated over large reaches?

More text has been added to the discussion to explain how f_s and f_c can be extrapolated / calculated, and how r can be estimated as a function of the slope and the discharge (lines 332-370).

Figure 1 – can't see the inset graphs and they don't have any units to even

know if the upstream and downstream concentrations are the same.

Figure 1 has been deeply revised, units have been added to the insets and the relationship between upstream and downstream concentrations has been clarified putting both lines in the same plot, as suggested by the Referee 2. Thank you.

Figure 2 – what is a police probe circle color? Perhaps you don't need to say the actual colors as you have them in the figure key.

Colors and color names have been changed, thank you.

Figure 3 – again the names of the label colors (blue petroleum) are not needed and are distracting - or just call it a simple name. A and C sort of say the same thing right - consider if you need both and are these actually referenced in the paper? They are only briefly discussed in lines 204-205, as such you might consider their importance to the audience.

Colors and color names have been changed. Panel a shows the pattern of r versus Q , while panel c indicates how frequently different values of Q and r were observed in the study reach during the focus season, an important information which is not provided by panel a. This was further clarified in the revised text (lines 267-276 and 283-287).

Please consider a map so we know where you are, and what this stream may look like within the southern Alps. Photos do not show the slope very well.

A new Figure in the SI has been added to show the catchment and the location of the Valfredda in the southern side of the Alps. thank you (Figure S8).

Please consider much of this to be worthy methods and material for discussion in a longer format journal.

While we understand the concern of the Referee, we also believe this is quite typical of papers published in multi-disciplinary journals with severe space constraints. While we agree that these details could be interesting, they could be perceived as distracting by researchers who are not familiar with the empirical measurement of the patterns of gas concentration in streams.

References

1. Barker, S. & Ridgwell, A. Ocean Acidification. Nature Education Knowledge **3** (10 2012).

2. Botter, G., Basso, S., Rodriguez-Iturbe, I. & Rinaldo, A. Resilience of river flow regimes. Proceedings of the National Academy of Sciences **110**, 12925–12930. ISSN: 00278424 (Aug. 2013).

Reviewer #1 (Remarks to the Author):

This manuscript is a revised version of one I reviewed earlier. This version is much easier to understand, thanks to rewriting, Fig 1 etc. I also like the scaling exercise where we can see how steps would increase scaled estimates of CO₂ fluxes (though I note from having seen 2 of these streams, they were plane bed and not step pool, but the point of the exercise is well stated). I have a few small comments to consider.

4 consider a sentence that states the current models of gas exchange include only things like slope, velocity depth and ignore withing reach spatial heterogeneity

5 replace signature with imprint.

108 located between

175. Scaling to year seems too long given that CO₂ was sampled only in one season, consider using daily units.

197 "This" refers to an unclear antecedent forcing the readers to guess at what "this" stands in for. Follow with a noun.

320 ditto

356-361. I like this simple exercise, it could be combined with the picture in SI 1.7 to form a stand alone box as it nicely encapsulates the main point that for 2 reaches with same slope and length, the one with steps will have higher gas exchange velocity.

374. novel is too simple a word here to describe the qualities of these scaling laws, they would be both more realistic and more complicated.

379. See this paper for the same point
<https://doi.org/10.31223/osf.io/8u6vc>

418 Makes me wish we had measured longitudinal bed profiles for reaches with tracer gas exchange measures. Yes in the future we should measure the morphology of these reaches in a more detailed way.

418 Delete careful here and in the other spot—no other way to do science.

548 Referring to the frequency distribution of the CO₂ traces with time makes me think that alternative way of analyzing these data would capitalize on the uncertainty, so rather than take the mean of the CO₂, consider it as a probability distribution and scale this uncertainty upward using hierarchical Bayes. I am not asking for wholesale reanalysis of the paper here, would take a long time! but rather pointing out that there is perhaps a better way to deal with the uncertainty in measuring CO₂, which the authors nicely show can be quite high.

Fig 2 is great

Reviewer #2 (Remarks to the Author):

I believe the authors have adequately met the concerns I raised in my initial review. Many of the technical aspects of the study design are clearer, and I appreciate the changes made to figures as well. The main issue with the original work was whether or not the author's claims could be appropriately extrapolated beyond the small study stream and limited temporal scope. I believe they have made improvements in demonstrating how their methods can be used more widely. This and the new text also helps in making their case to a less specialist audience as is more suitable to this journal.

line notes (track changes version):

32-34: a little hard to follow cause and effect through this sentence- are you trying to describe how the elevation change in the channel leads to the jet and the turbulence of the jet causes/has caused the pool to form leading to a deeper (use of "stage"?) stream cross section? or something else related to morphology and jet plunge location?

116: not sure what is meant by "without loss of generality"

451-455: Suggest restructuring sentence so the e.g. does not break it up and better explaining each part. 1) provide evidence of this preferential use of tracers and why 2) steps were disregarded because segments were assumed to be more important or because segments WERE more important but steps also need to be included? Clarify thought...

Reviewer #3 (Remarks to the Author):

Overall the major comments that I had have been addressed with the exception of one. As raised in the initial review, the manuscript does a great job of describing the hydrologic importance of their findings to the general considerations of gas exchange. Furthermore, the authors have provided a manuscript that is more readable generally, and key factors are better explained. The authors have also included an assessment comparison with a reference Horgby et al 2021, that further shows the potential impact that these findings may have on the emissions of CO₂ from a headwater environment. However, one concern still remains that the manuscript is focused so narrowly on the hydrologic components, that an opportunity is missed to provide the context for the importance of these findings to ecosystem dynamics as we currently measure them. This not only includes the scaling of CO₂ emissions, a topic that is given significant attention in the literature, but these findings could also suggest increased uncertainties with efforts to estimate stream metabolism based on oxygen differentials. As stated by the authors in their response, they do not believe that a discussion like this is warranted and that this should emphasize the broader application of hydrodynamics to gas exchange. However, if this was the goal from the beginning, it would have been prudent to utilize an inert gas, Argon, or something that is less affected by biology to emphasize the mass exchange. I am not suggesting that this precludes this from publication, but I suggest that the authors are missing an opportunity to make this work broader in its reach, especially if being considered for Nature Communications. Again, this is my only remaining concern is that the paper may be too narrow and that this contribution could be better suited for a more specialized journal for hydrologists specifically if that is their intent. If the editors are not concerned about this, or if other reviewers do not agree, I would support publication in Nature Communications as this is very well presented and written.

Response to the Referees

We would like to thank the reviewers for handling our work and providing insightful comments, which led to a considerable improvement of the first manuscript. In what follows, a point-by-point response to all the comments is presented. In this rebuttal, text taken from the decision letter is presented as blue indented text. Each comment is followed by a detailed response, with a list of actions implemented. Where specified, the line numbers refer to the manuscript version with tracked changes. The large majority of the comments received were properly complied with.

Reviewer 1

This manuscript is a revised version of one I reviewed earlier. This version is much easier to understand, thanks to rewriting, Fig 1 etc. I also like the scaling exercise where we can see how steps would increase scaled estimates of CO₂ fluxes (though I note from having seen 2 of these streams, they were plane bed and not step pool, but the point of the exercise is well stated).

We thank the reviewer for the positive assessment of our paper.

Minor comments

4 consider a sentence that states the current models of gas exchange include only things like slope, velocity depth and ignore withing reach spatial heterogeneity

Thanks for the suggestion. However, the abstract is limited to only 150 words and there is no space to add such a sentence.

5 replace signature with imprint.

Done, thanks (line 5).

108 located between

Fixed, thanks (line 111).

175. Scaling to year seems too long given that CO₂ was sampled only in one season, consider using daily units.

Thanks for the suggestion, the units have been changed (line 178).

197 “This” refers to an unclear antecedent forcing the readers to guess at what “this” stands in for. Follow with a noun.

Done, thanks (line 201).

320 ditto

Fixed, thanks (line 324).

356-361. I like this simple exercise, it could be combined with the picture in SI 1.7 to form a stand alone box as it nicely encapsulates the main point that for 2 reaches with same slope and length, the one with steps will have higher gas exchange velocity.

We agree with the comment. However, by looking at the author’s guide we don’t think it’s possible to do that in a paper.

374. novel is too simple a word here to describe the qualities of these scaling laws, they would be both more realistic and more complicated.

Thanks for the comment. We added the words ”more sophisticated” (line 389).

379. See this paper for the same point <https://doi.org/10.31223/osf.io/8u6vc>

Thanks for bringing this contribution to our attention. However, we noticed that this is a preprint that has not been subjected to any type of peer review. While by any means we are not saying that the work is either wrong nor not valuable, we think that the peer review process is a fundamental step of the scientific methodology that every research should comply with. Therefore, we decided not to cite the preprint paper in question.

418 Makes me wish we had measured longitudinal bed profiles for reaches with tracer gas exchange measures. Yes in the future we should measure the morphology of these reaches in a more detailed way.

Thanks for the comment, this is one of the main points of the paper.

418 Delete careful here and in the other spot—no other way to do science.

Done, thanks (line 435).

548 Referring to the frequency distribution of the CO₂ traces with time makes me think that alternative way of analyzing these data would capitalize on the uncertainty, so rather than take the mean of the CO₂, consider it as a probability distribution and scale this uncertainty upward using hierarchical Bayes. I am not asking for wholesale reanalysis of the paper here, would take a long time! but rather pointing out that there is perhaps a better way to deal with the uncertainty in measuring CO₂, which the

authors nicely show can be quite high.

Thank you, we added a sentence to clarify the nature of our analysis and highlight how this differ from a full uncertainty analysis (line S176).

Fig 2 is great

Thanks!

Reviewer 2

I believe the authors have adequately met the concerns I raised in my initial review. Many of the technical aspects of the study design are clearer, and I appreciate the changes made to figures as well. The main issue with the original work was whether or not the author's claims could be appropriately extrapolated beyond the small study stream and limited temporal scope. I believe they have made improvements in demonstrating how their methods can be used more widely. This and the new text also helps in making their case to a less specialist audience as is more suitable to this journal.

Thank you, we are glad to see that the changes we made led to an improvement of the manuscript.

Line notes

32-34: a little hard to follow cause and effect through this sentence- are you trying to describe how the elevation change in the channel leads to the jet and the turbulence of the jet causes/has caused the pool to form leading to a deeper (use of "stage"?) stream cross section? or something else related to morphology and jet plunge location?

Thanks for pointing this out. The sentence has been rewritten to be clearer (line 32-42).

116: not sure what is meant by "without loss of generality"

We removed the phrasing (line 115).

451-455: Suggest restructuring sentence so the e.g. does not break it up and better explaining each part. 1) provide evidence of this preferential use of tracers and why 2) steps were disregarded because segments were assumed to be more important or because segments WERE more important but steps also need to be included? Clarify thought...

Thanks for the comment. The sentence has been rephrased to solve these issues (line 399-407).

Reviewer 3

Overall the major comments that I had have been addressed with the exception of one. As raised in the initial review, the manuscript does a great job of describing the hydrologic importance of their findings to the general considerations of gas exchange. Furthermore, the authors have provided a manuscript that is more readable generally, and key factors are better explained. The authors have also included an assessment comparison with a reference Horgby et al 2021, that further shows the potential impact that these findings may have on the emissions of CO₂ from a headwater environment.

We thank the reviewer for helping us to improve the manuscript.

However, one concern still remains that the manuscript is focused so narrowly on the hydrologic components, that an opportunity is missed to provide the context for the importance of these findings to ecosystem dynamics as we currently measure them. This not only includes the scaling of CO₂ emissions, a topic that is given significant attention in the literature, but these findings could also suggest increased uncertainties with efforts to estimate stream metabolism based on oxygen differentials. As stated by the authors in their response, they do not believe that a discussion like this is warranted and that this should emphasize the broader application of hydrodynamics to gas exchange. However, if this was the goal from the beginning, it would have been prudent to utilize an inert gas, Argon, or something that is less affected by biology to emphasize the mass exchange. I am not suggesting that this precludes this from publication, but I suggest that the authors are missing an opportunity to make this work broader in its reach, especially if being considered for Nature Communications. Again, this is my only remaining concern is that the paper may be too narrow and that this contribution could be better suited for a more specialized journal for hydrologists specifically if that is their intent. If the editors are not concerned about this, or if other reviewers do not agree, I would support publication in Nature Communications as this is very well presented and written.

We added a sentence trying to address the implications of this study for the indirect estimation of stream metabolism based on oxygen differential, and further stressed the links of our work with the stream carbon cycling (lines 344-347 and 350-357).